# Reconciling metal–silicate partitioning and late accretion in the Earth

Terry-Ann Suer [1,2 ✉], Julien Siebert [3,4], Laurent Remusat[1], James M. D. Day [2,5], Stephan Borensztajn[3], Beatrice Doisneau[1] & Guillaume Fiquet[1]

Highly siderophile elements (HSE), including platinum, provide powerful geochemical tools for studying planet formation. Late accretion of chondritic components to Earth after core formation has been invoked as the main source of mantle HSE. However, core formation could also have contributed to the mantle's HSE content. Here we present measurements of platinum metal-silicate partitioning coefficients, obtained from laser-heated diamond anvil cell experiments, which demonstrate that platinum partitioning into metal is lower at high pressures and temperatures. Consequently, the mantle was likely enriched in platinum immediately following core-mantle differentiation. Core formation models that incorporate these results and simultaneously account for collateral geochemical constraints, lead to excess platinum in the mantle. A subsequent process such as iron exsolution or sulfide segregation is therefore required to remove excess platinum and to explain the mantle's modern HSE signature. A vestige of this platinum-enriched mantle can potentially account for [186]Os-enriched ocean island basalt lavas.

[1] Institut de Minéralogie, de Physique des Matériaux et de Cosmochimie, UMR CNRS 7590, Museum National d'Histoire Naturelle, Sorbonne Université, Paris, France. [2] Department of Earth and Planetary Science, Harvard University, Cambridge, MA, USA. [3] Institut de Physique du Globe de Paris, UMR CNRS 7154, Paris, France. [4] Institut Universitaire de France, Paris, France. [5] Scripps Institution of Oceanography, University of California San Diego, La Jolla, CA, USA. ✉email: terry_suer@fas.harvard.edu

The highly siderophile elements (HSE) comprise Re and Au along with the platinum-group elements (Os, Ir, Ru, Rh, Pt, and Pd). Due to their strong affinity for Fe-metal at low pressures (1 bar), these elements are stripped from silicate mantles during metallic core formation of differentiated planetary embryos and protoplanets. Earth's mantle is accordingly strongly depleted in the HSE with respect to chondrites, assumed to represent the materials that make up the bulk Earth, with more than 99% of the HSE residing in the core[1]. Yet the abundances of HSE in terrestrial rocks are far greater than predicted by metal–silicate partitioning experiments, a long-standing issue in geochemistry known as the "excess siderophile element problem"[2]. Partitioning data predict large differences in the relative mantle HSE abundances due to their differing affinities for the metal phase[3,4], while the terrestrial mantle displays broadly chondritic relative HSE abundances[5]. Furthermore, the long-lived $^{190}$Pt–$^{186}$Os isotope system indicates a chondritic evolution of the Pt/Os ratio of the bulk silicate Earth (BSE) through precise measurements of the $^{186}$Os/$^{188}$Os ratio in mantle rocks[6]. These observations indicate that the HSE abundances of the present mantle record the imprint of late accretion of a small amount of dominantly chondritic material (~0.3% to 1% $M_E$)[5], also referred to as the late veneer[7,8].

Non-chondritic HSE signatures have also been noted in Earth's mantle. For example, the suprachondritic Ru/Ir ratio[6] cannot be reproduced by accretion of any known chondrite groups. In addition, W isotopic heterogeneities with respect to chondrites and BSE are observed in some of the earliest known Archean rocks and could record the presence of a pre-late accretion mantle (e.g.,[9,10]). Depending on the mechanisms that were active, the early mantle could have already contained a portion of the total mantle HSE budget as suggested by the presence of positive and negative $\varepsilon^{182}$W anomalies[10–12]. Similarly, some terrestrial Archean samples show non-chondritic platinum stable isotope compositions, likely reflecting the imprint of core-forming processes[13]. These observations are consistent with an alternative hypothesis of core-mantle equilibration at high pressures and high temperatures (HP–HT) as a way of producing the excess of HSE in the mantle[14]. Recent studies carried out up to 18 GPa and 2773 K[4] provide evidence that metal–silicate partitioning of HSE are lowered by increasing P–T conditions and predict that core formation could account for mantle abundances of palladium and platinum[3,4,15]. The present mantle HSE signature could therefore reflect a combination of a number of processes including late accretion and core formation. However, the development of small size (ranging in size from ~50 nm to ~1 μm) metallic inclusions dispersed in quenched silicate experimental melt run products, often referred to as nuggets, has been a source of difficulty for the interpretation of HSE partitioning values from some of these studies (e.g.,[16–18]). If nuggets are present as equilibrium particles during melting, they are contaminants in the silicate and should not be included in the estimates for partition coefficients[18]. These analytical issues combined with low solubility of the HSE in silicate melts have hindered the acquisition of HSE partitioning data at higher P–T conditions directly relevant to the Earth's core formation at deep magma ocean conditions[19–21].

Platinum is a key element for solving the excess siderophile problem due to its strongly evolving partitioning behavior with P–T conditions[4,15] and its role as the long-lived (half life of $^{190}$Pt is 470 Ga) parent in the $^{190}$Pt–$^{186}$Os isotopic system[13]. In this study, we add constraints to the origin of the mantle's platinum content by providing the first measurements of metal–silicate partition coefficients of platinum carried out in the laser-heated diamond anvil cell (LHDAC) (Supplementary Table 1). This is the only static compression device capable of producing the

HP–HT conditions relevant to a deep magma ocean (e.g.,[22]). The samples were compressed to pressures between 43 and 111 GPa and heated to temperatures between 3600 and 4300 K for a few tens of seconds (see Methods). Cross-sections from the quenched molten region of the samples were recovered and prepared for chemical analysis using a focusedionbeam (FIB). A series of analytical instruments were then used to tackle the experimental difficulties related to the study of HSE partitioning. These include notably, the NanoSIMS (i.e., nanoscale secondary ion mass spectrometry) which provides suitable analytical and spatial resolution for measuring the expected low platinum content dissolved in the silicate melt of the run products at the submicron scale of the LHDAC quench product. In addition, we used transmission electron microscopy (TEM) to assess the presence of nanoparticles/nuggets and to evaluate how they formed in the LHDAC samples.

## Results

**Metal–silicate partitioning experiments.** The post-experimental run products consisted of a spherical metal blob (up to 15 microns in diameter) enveloped in a quenched molten silicate envelope of roughly 20–40 microns in diameter (Fig. 1). This geometry is typical of superliquidus partitioning experiments in which these two phases melted and equilibrated[21,23]. The major elemental chemical compositions of the two regions were measured by SEM-Energy Dispersive X-rays (EDX) and the platinum concentration in the silicate was quantified by NanoSIMS (compositions are reported in Supplementary Tables 1 and 4). Reported NanoSIMS concentrations in the silicate sections of the run products were averaged over several small regions, avoiding heterogeneities in the sample (see Fig. 1g and Supplementary Table 5). Small metallic particles (<100 nm) dispersed in the silicate melt were observed in electron images and further probed by TEM. EDX analysis performed with the TEM allowed us to quantify the composition of these metal nanoparticles present in the silicate melt of a representative sample. The Fe/Pt ratios of the particles were observed to be ~9 times higher than that of the central equilibrium metal blob (Fig. 1d, e and Supplementary Table 6) indicating that they were neither mechanically extracted from nor in equilibrium with the central metallic blob. The high average Fe/Pt ratios of the particles (~12.5) also suggest that they have negligible influence on the platinum concentration in the silicate. Based on this, we conclude that these particles were dissolved in the silicate melt at superliquidus conditions and exsolved upon rapid quench from high temperatures. This conclusion is similar to that of recent work which shows that metallic microspheres in the quench silicate from platinum partitioning experiments are mostly composed of iron and likely formed from a smelting process during quench[24]. Moreover, similar quench features in the silicate melt were previously observed in metal–silicate LHDAC experiments[21,23] and therefore cannot be attributed only to the presence of HSE.

**Core formation modeling.** During core-mantle equilibration, trace elements such as platinum are distributed between the metal and silicate phases depending upon their metal–silicate partition coefficient:

$$D_{Pt}^{met/sil} = \frac{X_{Pt}^{met}}{X_{PtO_{\frac{n}{2}}}^{sil}} \qquad (1)$$

Where $X$ is the molar fraction of platinum of the respective components. The measured partitioning coefficients from our experiments containing platinum-rich alloys were corrected to platinum concentration at the level of infinite dilution using a Margules mixing model for a binary solution[4,25]

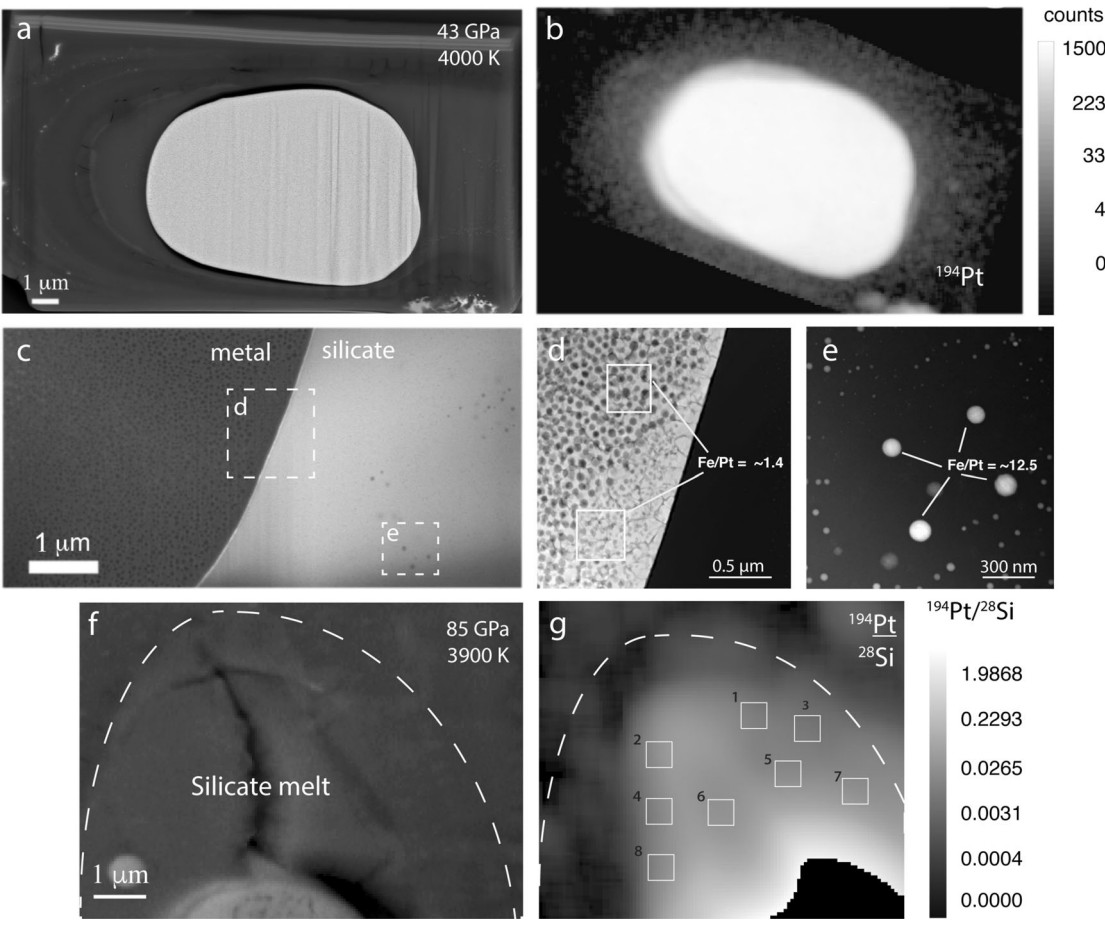

**Fig. 1 Characterization of experimental samples. a** Backscattered electron (BSE) image of run # 1 showing the main quenched metal. reservoir surrounded by quenched silicate melt pocket and an outer rim of unmelted silicate. **b** NanoSIMS $^{194}$Pt map of run # 1 showing the overall platinum distribution in the whole sample. **c** BSE image of run # 1 after thinning by focusedion beam (FIB) down to a thickness of ~100 nm. **d** An image taken by high-resolution transmission electron microscopy (HRTEM) of metal quench textures of the main metal blob of run # 1. The average Fe/Pt ratio of the main metal is ~1.4 (in weight % ratio) and was obtained by energy dispersive X-ray (EDX) analysis. **e** HRTEM image showing dispersed nanoscale metallic particles from run # 1 quench silicate. The nanoparticles have an average Fe/Pt ratio of ~12.5 (weight % ratios), (see Supplementary Table 6). **f** BSE image of run # 5 showing the region of contact between the quenched silicate and metal. **g** NanoSIMS $^{194}$Pt/$^{28}$Si map of the silicate portion of run # 5. The boxes show the analysis regions (see Supplementary Table 5). Metal blob is masked out in lower right corner due to its high Pt concentration.

(see Supplementary Notes 1). The infinite dilution corrected partition coefficients ($D_{Pt}^0$) reported in Fig. 2 are accordingly representative of core-forming alloys. At conditions of our experiments, the metal–silicate partitioning of platinum is lowered by a few orders of magnitude with respect to previous results obtained at lower P–T conditions[4,15,24,26]. However, there is an excellent agreement among these datasets as all of the partitioning data lie on a single linear trend as a function of temperature (Fig. 2). This result indicates consistency of our new platinum partitioning values obtained in LHDAC with those from previous studies carried out in large volume press (LVP)[4,15]. This is a further validation that metal–silicate partitioning experiments carried out in LHDAC are able to sample equilibrium chemistry and provide results that are consistent with those of lower P–T experiments. A multiple variable linear regression analysis was carried out on a dataset that combines some of the previous results with those of the current study[4,15,24,26]. This analysis showed that platinum has greatly reduced siderophile tendency with increasing temperature while the effect of pressure on its partitioning behavior is negligible, consistent with the conclusions reached by earlier studies[4,15]. The effect of oxygen fugacity ($fO_2$) was determined to be negligible over the range of the combined dataset, an indication that platinum partitioning is independent

of $fO_2$ below +2 of the iron–wüstite (IW) redox buffer, where the majority of these measurements were acquired. This conclusion is supported by recent work[24] showing that platinum can be dissolved as a neutral species in silicate melts at the $fO_2$ conditions relevant to the current study ($-1.25 <$ IW $< -0.67$). Finally, the significant amounts of light elements Si and O (see Supplementary Table 1) dissolved in the metal at conditions of our experiments do not affect platinum partitioning. The final parameterization determined for the infinite dilution corrected platinum metal–silicate partitioning, $D_{Pt}^0$ is expressed as a function of temperature:

$$\text{Log}\, D_{Pt}^0 = -2.99(\pm 0.24) + \frac{21817(\pm 516)}{T} \qquad (2)$$

This parameterization was applied to understanding the distribution of platinum between the mantle and the core during the Earth's growth through a suite of core formation models (see Methods) (e.g.,[27,28]). Platinum accumulates in the mantle as the Earth grows according to the change in its metal–silicate partitioning behavior with evolving P–T and redox conditions of core-mantle equilibration[21,29,30]. When considering only the P–T constraints for core-mantle equilibration at deep magma

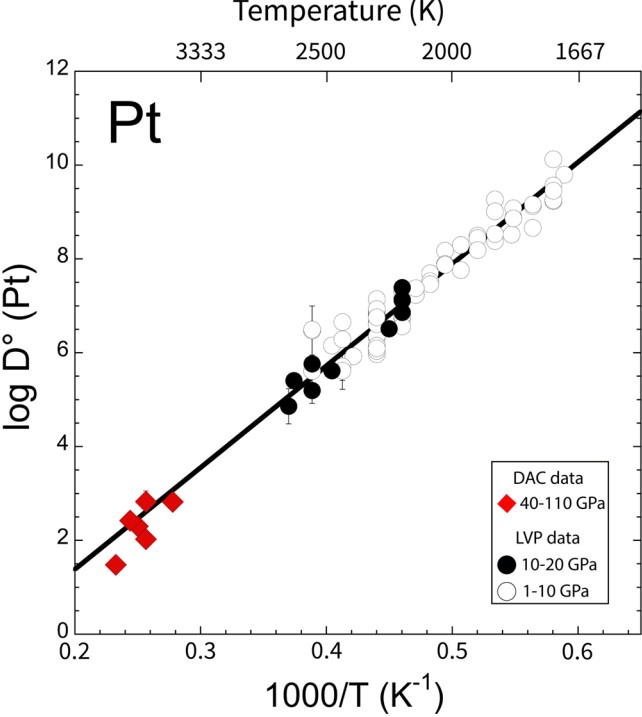

**Fig. 2 Platinum metal–silicate partitioning behavior.** Metal–silicate partitioning coefficients of platinum, $D^0$ (Pt) (corrected for infinite dilution) as a function of inverse temperatures. Red diamonds are the results of laser-heated diamond anvil cell (LHDAC) experiments from the current study (one sigma error bars fall inside plot symbols) at pressures between 40 and 110 GPa temperatures from 3600 to 4300 K. The black and white circles show the results of previous studies carried out at lower pressure-temperatures conditions in large volume press experiments[4,15,24,26]. The white circles represent data obtained from experiments carried out at pressures from 1 to 10 GPa while the black circles are data from 10 to 20 GPa. The combined dataset is fitted to Eq. (2). by multiple variable regression analysis. The best fitting function for the data, shown by the solid black line, is strongly dependent on temperature.

ocean conditions (i.e., $45\,\text{GPa} \geq P \leq 65\,\text{GPa}$ and $T \geq 3500$ K)[20,21,29], the model can account for the platinum content of the BSE [$8.6\,\text{ppb} \pm 1.7$[6]], without the requirement for late accretion (Fig. 3a, black line). Within an acceptable P–T range, this result is also compatible with the addition of ~0.38% $M_E$ late accreted material comprised of carbonaceous chondrite (CI) (Fig. 3a, blue line), a value that falls within the range of late accretion mass estimates of ~0.3 to 1% $M_E$[2,5]. In this scenario, the platinum content of the present mantle could be explained by a hybrid model in which platinum and HSE abundances in the BSE are the combined result of both metal–silicate partitioning as well as an overprint of late accretion. This supports the conclusions of some previous works conducted at lower P–T conditions[3,4,15,31]. However, above ~55 GPa, core-mantle equilibration leads to a mantle platinum content incompatible with the minimum mass of late accretion required to account for the abundances of other HSE (Fig. 3a, blue horizontal dashed line). This hybrid model is therefore too restrictive. Additionally, existing partitioning data at HP–HT strongly suggests that core-mantle equilibrium cannot generate the broadly chondritic relative proportions of all HSE (e.g., Os, Ir, Re, and Pd) in the BSE[4,31].

A more comprehensive set of core formation modeling that incorporated our new partitioning result considered the effect of a broader range of parameters on the mantle's platinum content (see Methods). These include the mantle initial and final redox state, mantle abundances of moderately siderophile elements (Ni–Co–V–Cr)[21,30], seismologically consistent final core compositions[32] and a range of magma ocean geotherms[22,33]. All results from these models producing both the siderophile element abundances of the BSE and a core composition in light elements consistent with the geophysical constraints[32] lead to an excess of platinum relative to the present BSE, even before late accretion (black line Fig. 3b). Partial equilibration rather than full equilibration of metal and silicate only shifts the solution space of core-mantle equilibration to higher P–T conditions in order to account for the abundances of the moderately siderophile elements (see Methods).

## Discussion

These combined results suggest that core formation in a deep magma ocean[20,21,30] would lead to an overabundance of platinum in the mantle relative to modern observations[6]. However, the chondritic $^{186}Os/^{188}Os$ and $^{187}Os/^{188}Os$ ratio strongly implies a chondritic late accretion component delivered a significant portion of the mantle's complement of the HSE[6]. If the platinum content of the mantle is largely due to late accretion, then a significant fraction of the platinum from core formation must have been lost prior to this process. Fractionation processes in a magma ocean must therefore be responsible for ultimately lowering high concentrations of platinum and the other HSE to explain their present-day levels. Our results provide indirect evidence that self-reducing processes such as sulfide exsolution[34] and/or iron disproportionation[35] operated in the early Earth and were responsible for sequestering the excess of platinum and other HSE leftover from core formation. Recent work has shown that the disproportionation of $Fe^{2+}$ to $Fe^{3+}$ and $Fe^0$ in a magma ocean could have produced a lower bound of between 0.1 and 3 wt.% of metallic iron[36]. This amount of precipitated Fe ($10^{21}–10^{23}$ kg, equivalent to the core mass of a planetesimal or planetary embryo) could sequester the excess of platinum (as high as few tens of ppb) that accumulated in the mantle from core formation (Fig. 4a). Unlike the cores of impacting planetesimals or embryos, the precipitated Fe droplets would be free of impurities and dispersed in the magma ocean. If efficient metal–silicate equilibration occurs this would then allow the Fe droplets to sequester platinum and other HSE, from a range of depths, eventually segregating (by diapirism or percolation) to merge with the core[37]. Our HP–HT results therefore support the requirement of such a mechanism in the early Earth's magma oceans, that can also explain the rapid increase in oxidation state of the mantle from the reduced conditions of core forming ($fO_2$ ~IW) to the more oxidized modern values (fayalite-magnetite-quartz—FMQ)[38].

The amount of precipitated Fe required to segregate HSEs from a magma ocean could be used to place additional constraints on core formation and the post core formation state of the mantle. Armstrong et al.[36] showed that if as little as 0.1 to 3 wt.% of precipitated Fe segregated from a magma ocean, the resulting $Fe^{3+}/\sum Fe$ ratio would match that of the modern-day mantle. As described above, this amount Fe could remove the excess Pt leftover by core formation before segregating to merge with the core. Metal–silicate partitioning measurements for the other HSE, similar to the ones we have obtained for platinum would help to constrain the lower bound of precipitated Fe required to both sequester HSE and to account for the mantle's current redox state. Further LHDAC measurements will therefore clarify the conditions that existed in the magma ocean during and immediately after the Earth's core formed.

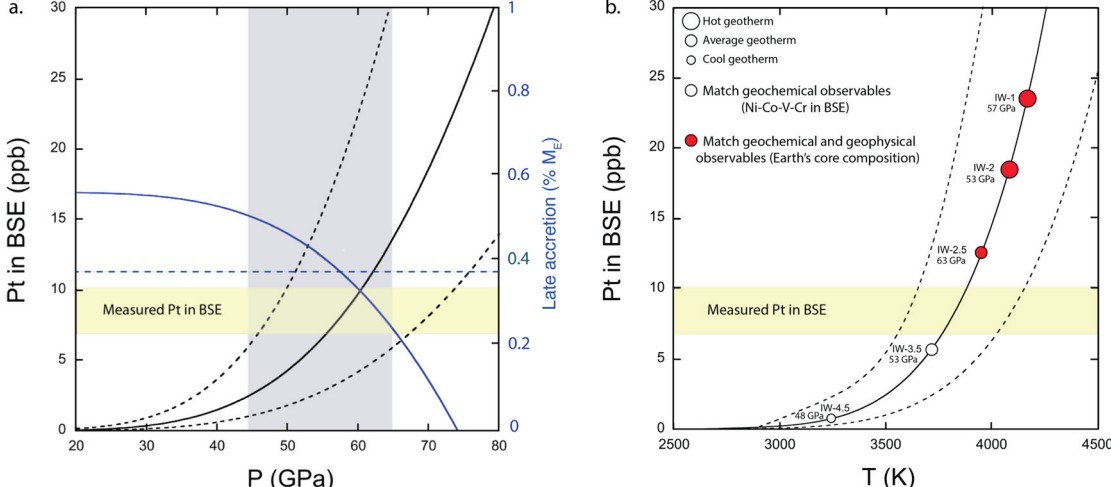

**Fig. 3 Results of core formation modeling. a** The solid and dashed black curves are the average and the uncertainty envelope of the modeled platinum in the bulk silicate earth (BSE) against final equilibration pressure (i.e., each pressure corresponds to the last pressure of equilibration of a continuous core formation process during accretion). This is the result of core formation model in which the temperature follows the liquidus temperature of a pyrolite type compound averaged from the experimental studies[22,33] (see section "Core formation and accretion modeling" of Methods for further). The yellow rectangle show the range of platinum measured in the mantle[6]; the gray rectangle shows the pressure constraints on core formation conditions from previous studies[21,29]. At each pressure, the blue curve corresponds to the maximum amount of late accretion material that can be added to the lower bound of platinum leftover from core formation and not exceed the observed amount of platinum measured in the present mantle. Horizontal blue line is the minimum mass of carbonaceous chondrites (CI) late accretion necessary to explain the other highly siderophile elements (HSE)[2]. **b** The solid and dashed lines represent the average and uncertainty envelope of the platinum in the mantle against final equilibration temperatures of a continuous core formation process during accretion. Circles show the final platinum content from a selection of models consistent with geochemical constraints and light element core contents. Three different geotherms are used in the modeling as denoted by the sizes of the circles (where cooler geotherms are smaller sized and hotter geotherms are larger sized circles), and five different initial redox conditions for Earth's accretion and core formation are considered. Displayed pressures represent final core-mantle equilibration pressures. In all cases, the final oxygen fugacity (fO$_2$) of the models is fixed by the current fO$_2$ content of the present core-mantle system (i.e., Iron–Wüstite –2.3) (see Methods for details). Model uncertainties are based on the uncertainty from the regression model of the partitioning data.

Our results further indicate that distinct early Earth reservoirs that avoided later iron disproportionation, such as those held responsible for preserving W isotopic heterogeneities[9–12], could also have enhanced HSE abundances after core formation. Metal–silicate equilibration in a deep magma ocean would have led to high Pt/Os ratios in resultant silicates. Inter-element fractionation is evident based on the measured platinum metal–silicate partition coefficients, and extrapolation of osmium metal–silicate partitioning behavior[31]. If a vestige of this early reservoir has been preserved since core formation in the first ~30 Ma of Solar System formation, it would have the combined characteristics of platinum concentrations similar to primitive mantle, and $^{186}$Os/$^{188}$Os that is 5–8% higher than in the present-day bulk silicate Earth[6]. Modern Hawaiian magmas have $^{186}$Os/$^{188}$Os higher than the ambient upper mantle[39]. Incorporation of between 5 and 10% of a primitive equilibrated reservoir into the sources of some these magmas[39] would explain radiogenic $^{186}$Os/$^{188}$Os measured within them, without collateral effects of enhanced HSE concentrations (Fig. 4b). Such a reservoir may also be associated with other early differentiation signatures (e.g., W, Ru, Mo anomalies, and high $^{3}$He/$^{4}$He)[10,40,41]. Constraining the partitioning behaviors of the other HSE (particularly osmium) at pressures and temperatures similar to those of the current work could thus allow further testing of the existence of an early equilibrated mantle reservoir.

## Methods
**Starting material.** The starting silicate used in this study was a natural Mid Ocean Ridge Basalt (MORB) from the East Pacific Rise which was micro-machined into small discs. Its composition is reported in Supplementary Table 1. A basaltic composition was used as a proxy for an average magma ocean composition due to its lower melting curves relative to peridotite or pyrolitic glasses[42]. Basalts melt at

lower temperatures and produce homogeneous quench silicate. Additionally, previous works on platinum partitioning did not report a significant effect of silicate composition (e.g.,[15]).

The (Fe, Pt) alloy was synthesized using a piston cylinder apparatus. The experiment was conducted at superliquidus conditions to segregate an Fe–Pt metallic blob from a basaltic glass. High purity metallic powders of Fe and Pt were mixed in a 1:2 ratio with powdered natural MORB. Synthesis was carried out at 15 kbars and 1800 °C using a standard ½″ BaCO$_3$ pressure cell assembly, with a graphite furnace and a MgO capsule. The recovered metallic blob of about 1 mm diameter was polished and analyzed with an SEM-EDX at 15 keV. The composition, averaged over several regions, showed that the blob was roughly 50: 50 wt.% Fe: Pt. Quench textures in the alloy indicated heterogeneities and it contained small amounts of other constituents such as S. Small pieces of metal were scratched from the bulk alloy and grounded down in an agate mortar and then pressed into foils of about 10 μm in thickness for loading into diamond cells.

**Laser-heating diamond anvil cell experiments.** For each experiment, a foil of the Fe-Pt alloy was sandwiched between two basaltic discs of 20 μm in thickness and placed inside a rhenium sample chamber. Gaskets were pre-indented to a thickness of ~30 μm and sample chambers of ~80 μm in diameter were made by a laser drill. The sample assembly was then compressed between two diamonds (culet diameters of 200 or 300 μm) in order to increase pressure in the sample chamber. A small ruby ball placed to one edge of each experimental chamber, away from the heated zone to avoid Al$_2$O$_3$ contamination of the silicate melt, was used as a pressure indicator by ruby fluorescence[43]. The shift in the diamond Raman peak (1334 cm$^{-1}$ at 1 bar) was also used to check pressures. A double-sided laser-heating system with a 200 W infrared laser ($\lambda_{max}$ = 1070 nm) and spot size 10–20 μm in diameter, was focused on the region of contact between metal and silicate and used to heat up and melt the sample.

The thermal emission spectra were collected by a Cassegrain-type objective with no chromatic aberrations. These spectra were collected simultaneously from each side of the DAC about every 2 s during the experiments. They were analyzed using a single-stage monochromator with a CCD detector. Temperatures were determined during the experiment by fitting the thermal radiation spectrum to a Planck function in the ~450–750 nm range[44]. Measured temperatures were the average of the central 5 microns of the hotspot. With emissivity assumed to be independent of wavelength in the Planck radiation function, temperature uncertainties are of the order of ±200 K. Temperature gradients were present

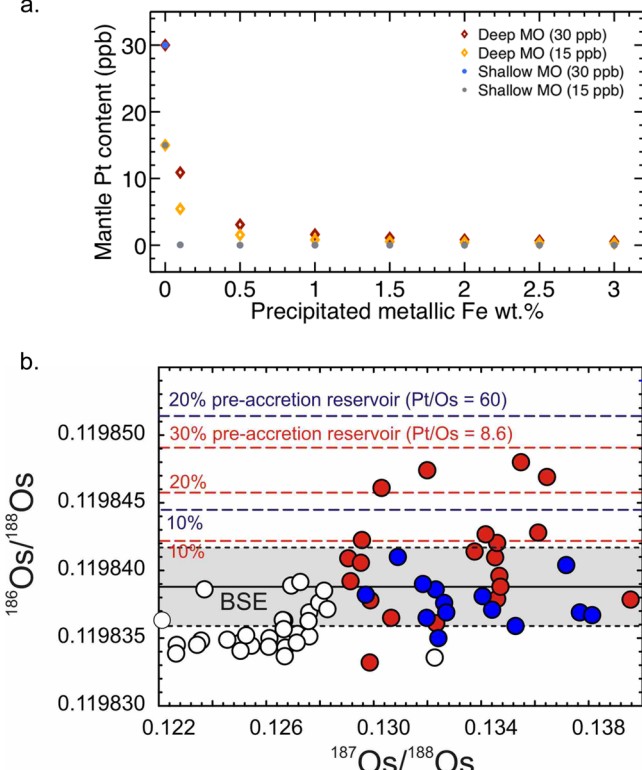

**Fig. 4 Mantle platinum content and osmium isotopic ratios. a** The platinum content in the mantle after removal by Fe precipitated from the disproportionation of $Fe^{2+}$ [35,36]. Both deep (diamonds) and shallow (dots) magma ocean (MO) with either 30 or 15 ppb of platinum after core formation are considered. If mixing is efficient, as little as 0.5 wt.% of Fe, in shallow (~25 GPa) and deep (~65 GPa) magma oceans can lower platinum in the mantle from several tens to a few ppbs. More Fe would be required in the case of less efficient mixing and segregation (more details in Methods). **b** Osmium isotope variations measured for abyssal peridotites (unfilled circles), lavas from Iceland (dark blue circles) and Hawaii (red circles) relative to the bulk silicate Earth (BSE) composition (gray shaded box)[6,39,72]. Hawaiian lavas with anomalous $^{186}Os/^{188}Os$ can be explained by a 10–20% contribution from an isolated deep mantle reservoir that did not experience disproportionation as shown by dashed red (Pt/Os = 8.6) and blue (Pt/Os = 60) lines, assuming mantle Pt = 30 ppb.

during LHDAC experiments. We did not have the ability (4-color imaging system) to quantify this gradient but these variations are instead considered in the uncertainties for the temperature measurements which incorporate the difference between the measured temperatures on the two sides of the sample, an analytical uncertainty of 100 K[45], and an uncertainty associated with the correction for axial temperature gradients[46]. The samples consist of a thin foil of (Fe, Pt) alloy sandwiched between two disks of silicate glass in controlled geometry. Temperature gradients are strong across a very thin boundary layer between partially molten sample and the cold surrounding glassy material, which acts as chemical and thermal insulation. Axial temperature gradients cannot be avoided within the heated area but are strongly minimized due to the superliquidus conditions of experiments[21,29,47].

The reported temperatures correspond to the highest average temperatures achieved during melting from both sides (e.g., Supplementary Fig. 1). The experiments were quenched rapidly at the highest temperatures by shutting off the laser power. The hotspot of each sample remained at superliquidus conditions for up to a minute ensuring chemical equilibrium between the metal and silicate phases. Previous works have determined that the timescale for equilibration during metal–silicate partitioning experiments at much lower temperature for larger samples is on the order of a few tens of seconds[29,48]. The extreme temperatures of LHDAC experiments at superliquidus conditions strongly favor chemical diffusion. For instance[49], estimated a diffusion length of ~40 μm in 1 s in silicate melt for Fe. The radii of the melt pockets in our experiments are around 15–20 μm. It has been shown in numerous previous LHDAC works[20,21,29,30,47,49,50] that chemical equilibration was reached in few seconds. This is well confirmed by the chemical

homogeneity of metal and silicate phases in run products of these experiments (where final T was kept for more than 10 s in all experiments).

Pressures were re-measured after the quench and a thermal pressure correction was applied to the ruby and Raman pressure estimates[51]. Pressure uncertainties reflect the difference in the preheated and postheated pressures. Experiments were carried out at pressures from 43 to 111 GPa and temperatures from 3600 to 4300 K. Experiments carried out at lower P–T conditions were unsuccessful due to unstable heating and the lower sensitivity of the CCD detector at low temperatures.

**Post-run sample preparation and analysis**. After decompression, a lamella of dimensions ~5 × 20 × 30 μm was recovered from the region of each sample that had been molten by use of a Zeiss Cross-beam focused-ion beam (FIB) instrument. A Ga+ beam operating at 30 kV was used to obtain a cross-section of the quench melt zone. Each surface of the lamellae were cleaned at a current of 2 nA and 200 pA for final surface polishing. Each lamella was attached by one corner to the tip of a tungsten micromanipulator needle and then placed flat on a silicon wafer using an electron curing adhesive from Kleindiek Nanotechnik (see Fig. S2). This geometry allowed for transfer to other analytical instruments.

Accurate NanoSIMS analyses of the samples required placement on a flat substrate in order to avoid signal artifacts from edge effects (observed when sections were welded to a Cu grid). Platinum deposition which is frequently used to protect underlying material during focused-ion milling was not used on these samples. The sections were ion polished on both sides and were of equal thickness across the whole section before being deposited flat on the wafer and attached with an electron curing glue. Once placed on the silicon wafer, the corners of the samples attached to the micromanipulator were cut by ion milling. The flatness of the sample was evidenced by further SEM observations before NanoSIMS measurements. The surface of each sample was cleaned again at a low current (100–200 pA) to remove any contamination prior to NanoSIMS measurements. After NanoSIMS analysis of the samples, a second round of FIB was performed on a section first extracted from run #1. This new section was welded to a copper grid and polished down at low Ga+ beam current to a thickness below 100 nm to allow observation and characterization at higher resolution of the nanostructures of the quench sample with TEM.

Backscattered electron images of the samples show typical textures of quenched liquids for both metal and silicate phases (Fig. 1). The general aspects of run products were very similar to previously described quench metal–silicate partitioning experiments performed with the LHDAC[21,23,30,47]. Metallic liquids exhibit heterogeneous texture due to the presence of a Fe-Si-O-Pt rich phases that exsolved during quench (Fig. 1). Such features in metal have often been observed in both large volume press[52] and DAC[21,30,53] partitioning experiments. The small sizes (<200 nm) of these exsolutions make them difficult to accurately characterize. Silicate textures (discussed more below under TEM section) typical of superliquidus experiments were also observed.

*Major element analyses*. The major element composition of the samples was characterized using energy dispersive X-ray analysis with a Zeiss Cross-beam field emission scanning electron microscope operated at 15 kV. Each sample was coated with a thin layer of carbon (~20 nm) before analysis. Metal and silicate phases of the run products were large enough (≥few μm) to obtain reliable EDX analyses. Moreover, large thickness of the FIB sections (≥3 μm) along with sample geometry of both phases made these samples suitable for EDX analysis. Several EDX spectra from silicate and metal phases were recorded during 60 s integrations and quantified with standards. The average compositions of metal and silicate and uncertainties are reported in Supplementary Tables 1, S4. Silicate and metal phases of each experiment are homogeneous at the scale of EDX analyses indicating that chemical equilibrium was achieved at conditions of our experiments. The quenched silicate is typical in composition to that reported from previous experiments on melting of basaltic glasses carried out in LHDAC, being similarly enriched in FeO with respect to the starting silicate material[29,42]. This FeO-enrichment is likely due to FeO partitioning into the melt. Melt/solid silicate partitioning experiments report values from 2 to 3 for Fe partitioning in favor of the melt (e.g., melt/Mg-perovskite partitioning experiments[49], similar to FeO-enrichment ratios observed in our experiments. This led to the redox conditions of the experiments being more oxidizing than expected (between ΔIW-1.26 and ΔIW-0.67). The quench metal consisted of iron ranging between 30.45 and 71.68 wt.% while the platinum content varied from 11.1 to 59.92 wt.%. Light elements silicon (0.85 to 3.51 wt.%), oxygen (4.34 to 6.86 wt.%), sulfur (0.64 to 8.4 wt.%), and traces of Mn, Al, Mg, Ti, Na, were also present in the metal. Variable amounts of S in the metal phase originated from the heterogeneous S contents of the starting (Fe, Pt) alloy, natural MORB also contains a few thousand ppm S. The potential presence of carbon in (Fe, Pt) alloys of the LH-DAC run products was assessed. Similarly to previous partitioning experiments[30] measured carbon contents on samples from a similar suite of experiments were below the detection limits (estimated around 1 wt.%)[29].

*Platinum analyses with the NanoSIMS*. The platinum concentration of the silicate region of each run product was measured by a CAMECA NanoSIMS 50 at the Museum National d'Histoire Naturelle in Paris. A 16 keV Cs+ beam was used to determine Pt contents through secondary ions of $^{28}Si^-$, $^{194}Pt^-$ and $^{27}Al^{16}O^-$. The

finely focused primary beam (with current of 23–26 pA) removed the top layers of sample material (from areas with field of views ~9 × 9 μm² (high-resolution maps) or 30 × 30 μm² (lower resolution maps)) to produce secondary ions which were then analyzed by a high-resolution double focusing multi-collection mass spectrometer[54]. Prior to each analysis, a high current Cs⁺ beam was used to pre-sputter large regions of the samples for up to 10 min. Integration times ranged from 30 to 90 min during which several tens of data frames (up to 60), which recorded accumulated counts, were collected. The data frames were deadtime corrected and aligned to produce a map of the total counts over all cycles, for each species. Precise rastering of the sample surface produced ion maps with resolution of ~300 nm. The spatial resolution of the maps depends on several factors including the ionization energy of the species being measured, the species concentration and the beam current (e.g.,[55]). Due to platinum's relatively high ionization energy and low abundance, a high primary beam current was necessary and this degraded the spatial resolution of the measurements compared to previous NanoSIMS measurements of sulfur in similar samples[23]. The NanoSIMS maps also appear slightly distorted with respect to electron images.

In order to determine platinum concentrations in the quench silicate from the NanoSIMS measurements, standard calibrations were carried out on silicate standards during each session. The standards were quantified a priori by laser ablation ICPMS (see Standardization subsection below) and a calibration curve obtained from a regression analysis of the standard measurements (Supplementary Fig. 3). An inverse regression following the method of[56] was used to determine the platinum concentration in the quench silicate of each run. This prediction model incorporates large extrapolations in concentrations between standards and samples in the prediction envelopes. The high-resolution of the ion maps (~300 nm) allowed platinum concentration to be measured in localized areas thereby avoiding contaminations from artifacts or discontinuities seen in the electron images. Contamination from subsurface artifacts were also avoided as only the top 100 nm of the samples are measured by NanoSIMS. The platinum concentrations in the silicate parts of each sample were averaged from four to eight regions of interests (ROIs) ranging from 0.5 to 1 μm² in size. The errors reported for the platinum concentrations in the silicates are based on the standard deviations of the ROIs.

*Standardization.* Certified NIST standards (NIST 612 and 610)[57] were used as silicate standards for compositional analysis. In-house silicate glass standards were also synthesized by adding trace amounts of platinum to natural MORB (NMORB) or synthetic MORB (FMORB). Homogenous mixtures were prepared in an agate mortar and melted with either a hydrodynamic gas laser levitation device at temperatures between 1873 and 2273 K or in a convection furnace at temperatures up to 1973 K. The quenched products of these syntheses were recovered, mounted in epoxy and polished for compositional analyses. Major element compositions were obtained by a Cameca SX100 microprobe. Typical operating conditions included an accelerating voltage of 15 keV with a 10 nA beam current. Integration times ranged from 10 to 60 s with averages made over several integrations. Some of the standards used for major elements were diopside (Si, Mg Ca), Fe₂O₃ (Fe), orthoclase (K, Al), albite (Na), MnTiO₃ (Ti, Mn), and Cr₂O₃ (Cr).

Trace element analyses of silicate standards were conducted with a laser ablation-inductively coupled plasma mass spectrometer (LA-ICPMS) at the University of Nantes. A Nd-YAG laser operating at 213 nm in pulse mode was used to ablate the samples. A Thermo Scientific Element XR inductively coupled plasma mass spectrometer (ICP-MS) with high elemental sensitivity and precision was used for elemental analysis. Operating conditions during the analysis included a beam diameter of 85 μm, laser frequency of 5 Hz and an energy of 83 mJ. Isotopes ¹⁹⁵Pt and ¹⁹⁴Pt were measured (NIST 612 was used as a standard for these measurements). The homogeneity of platinum in the silicate standards was verified at the scale of the LA-ICPMS and the NanoSIMS measurement footprints. The samples used as standards did not show spikes in their platinum concentration along profiles, which evidenced the absence of nano-nugget inclusions. Supplementary Table 2 summarizes the platinum contents of the silicate standards as measured by LA-ICPMS. Supplementary Figure 3 shows the platinum contents in the silicate glass standards measured by both LA-ICPMS and NanoSIMS, and a model fit to the data. Note that silicate standards containing high Pt content are difficult to synthesize due to Pt exsolution and nugget formation at the P–T conditions of the standard syntheses. Standards used contained ~0–16.38 ppm of Pt.

*Transmission electron microscopy (TEM).* A JEOL 2100F TEM operating at accelerating voltage of 200 KeV was used to analyze a thin section of run #1 (Fig. 1c–e). The lamella was probed with high angular dark field imaging to investigate quench textures in both the metal and silicate. The main metallic blob showed quench exsolutions (Fig. 1c, d) most likely due to the presence of O and Si in the metal at conditions of the experiments. Such metallic quench features have been observed in other LH-DAC partitioning experiments[21,29,30]. We also observed small metallic particles (50–500 nm) dispersed in the quench silicate melt. The general aspect and distribution of these particles are also very similar to those observed in previous LH-DAC partitioning experiments (e.g.,[21,29]) and consequently cannot be attributed only to the presence of HSE. As with those works, we interpret these particles as being due to exsolution during quench and not as equilibrium platinum nanonuggets.

This interpretation is sustained by EDX and electron diffraction measurements performed on these small metallic blobs in the TEM. Though it was difficult to assign a space group to diffraction patterns obtained on these inclusions due to their small sizes, the diffraction obtained from the main metallic blob and the small metallic inclusions indicate two different structures. Moreover, the EDX analyses performed in the TEM on the small metallic particles show very different compositions (Fe/Pt ~ 12.5) from that of the central main metallic melt (Fe/Pt ~ 1.4) (Fig. 1e and Supplementary Table 6) which would not be the case if the particles were were in equilibrium with the main metallic blob. These observations strongly suggest that the nanoparticles are most likely the result of the quench from high temperatures. These particles are also irregularly spaced in the samples and seem to form further than several hundred nanometers away from the main metallic blob. The NanoSIMS platinum analyses do not show a decrease in platinum concentration in the silicate closer to the main metallic blob. As such, these observations are interpreted as a further indication that the nanoparticles are quench features.

**Parameterizing and fitting.** Metal–silicate partitioning coefficients, D, describe the relative proportion of an element in a metal and a silicate phase during an equilibrium partitioning reaction (see main text Eq. (1)). Metal–silicate partitioning behavior of platinum can be parameterized in terms of quantifiable variables temperature (T), pressure (P), and oxygen fugacity (expressed here in terms of the IW buffer) and light element concentration of the metal: $\text{Log}D_{Pt}^{met/sil} = a + b + c\frac{P}{T} + d\Delta IW + e\text{Log}(1 - X_S) + f\text{Log}(1 - X_O)$ (e.g.,[15]). The valence of platinum in the silicate melt is known to depend on the oxygen fugacity conditions of the experiment[24]. We assumed a zero-valence state for Pt in the silicate melt in agreement with recent works which showed that under oxygen fugacities similar to the current experiments (ΔIW −1.26 to −0.67), platinum is likely to be present in the silicate melt in the zero-valence state[15,24]. Moreover, the present Pt partitioning dataset does not show any resolvable dependency with fO₂ of experiments in agreement with a zero-valence state for Pt in the silicate melt. It has been suggested that Pt can form anionic species such as PtC in silicate melts under similar redox conditions[15] from experiments done in LVP in which samples were encapsulated within graphite capsules. Though our analysis cannot rule out the presence of PtC and other anionic species in the quench silicate, there was no indication from the TEM EDX measurements to support the presence of anionic species.

Multiple linear regression analysis was conducted on a dataset which combines the partition coefficients measured in this study with those from previous works[4,15,24,26]. The dataset was fitted to the parameterization above to determine the constants a, b, c, d, etc. Some terms were not statistically significant and were removed and the data re-fitted until only statistically significant terms remained. The temperature was found to be the only statistically significant variable (i.e., pressure, metallic compositions and fO₂ have non-resolvable effects on Pt partitioning). We conclude that the temperature therefore exerts the dominant effect on platinum partitioning behavior over the range of the combined dataset (1–111 GPa, 1700–4300 K). This agreement strongly indicates that our results are consistent with the measurements obtained from LVP experiments, even though the measurement conditions do not directly overlap. This result also shows that partitioning measurements from this work are thermodynamically consistent with previous works arguing for chemical equilibration and relevant Pt measurements from current LHDAC experiments.

At conditions of our experiments (i.e., high temperatures), there are significant amounts of light elements (i.e., O, Si, and S) in the metal. Through the parameterization, we modeled the potential effects of metallic composition (presence of light elements) on the Pt partitioning. However, as reported in the manuscript, the effects of O, Si, and S are found to be statistically negligible on Pt partitioning. Recent work[58] reported a resolvable negative effect of sulfur on Pt partitioning. This effect was inferred from experiments containing large amounts of S in the (Fe, S) alloys (i.e., S contents above the eutectic composition in the Fe–S system and even stoichiometric Fe–S liquids). At conditions of our experiments, the sulfur contents of metal are relatively small (~0.1 for four experiments and two experiments where the sulfur contents are very low below 0.03 (i.e., <2 wt.%)). A significant effect of sulfur on Pt partitioning, particularly at the low sulfur compositions in our DAC data is not expected[58].

*Core formation and accretion modeling.* This new parameterized Pt partitioning was injected into continuous and multistage core formation models in which the Earth grows from the accretion of planetesimals and planetary embryos. Melting of the proto mantle occurs from the impact heating and release of gravitational energy. As the planet grows, the depth of melting increases and the P–T conditions of core-mantle equilibration increase along the peridotite solidus–liquidus. The cores of impacting planetesimals disperse and equilibrate in the silicate terrestrial magma ocean before merging with the Earth's core. Parameterized models of core formation[27] have been extensively used and upgraded in several recent works (e.g.,[23,28,29,32,59–61]).

In our models, accretion is discretized in a large number of steps. The first 80% of Earth's mass is added from 0.1% constant mass increments. The last 20% of accretion is dominated by larger impacts (2% mass increment) and a final 10% mass increment (representing the moon forming impact). Such growth scenarios have been suggested by dynamical simulations[8]. Mass (M) increases iteratively in

(i) steps (between few hundreds and 1000 steps depending on the fraction of large impacts considered in the models), each adding ($\delta M_i$) of the total Earth mass: $M_i = M_{i-1} + \delta M_i$ (Eq. S1).

All accreting planetesimals and embryos were differentiated; core and mantle of embryos were fully equilibrated at P–T conditions calculated as a function of their mass. Bulk compositions were assumed to be chondritic over the accretion history and the bulk composition of the growing Earth was equivalent to an assumed bulk Earth composition[62].

The final depth of the magma ocean is an important parameter of the models. To assess the effect of this parameter, the models consider a range of depths between shallow and deep magma oceans, which in terms of pressure, ranges from 0 to 135 GPa (present-day CMB depth). Since Pt partitioning is strongly temperature dependent, we also considered three different geotherms to test the effect of temperature on the final Pt composition of the BSE:

$$T = 1621 + 38.415P - 0.1958P^2 + 3.8369.10^{-4}P^3 \quad (S2)$$

$$T = 2022 + 54.21P + 0.34P^2 + 9.0747.10^{-4}P^3 \quad (S3)$$

$$T = 1/2 \left(2022 + 54.21P + 0.34P^2 + 9.0747.10^{-4}P^3 + 1940 (P/29 + 1)^{1/1.9}\right) \quad (S4)$$

Equation (S2) corresponds to the peridotite solidus following[22,33]. Equation (S3) is the hot liquidus following[22] while equation (S4) is an intermediate liquidus which corresponds to the arithmetic average of S2 and S3.

The concentrations of platinum and other elements in the resulting mantle and core were calculated based on chemical mass balance expressions for the accreting Earth, together with the partitioning parameterizations, determined in this study (Eq. 2) and those from the literature[29,30,32] for moderately (Ni, Co) and slightly (Cr, V) siderophile elements and the core composition in Si and O. We tested different starting redox conditions for the Earth ranging from very reduced (ΔIW-4.5) to oxidized (ΔIW-1). This covers the range of proposed starting redox conditions for Earth's accretion and differentiation[27,28,30,32,63]. The fO2 in these models was set to evolve to match the final fO2 of the present core-mantle system (ΔIW-2.3). An accreting embryo's mass balance in an element, $c$, is defined as:

$$c_b = F c_{ce} + (1 - F) c_{me} \quad (S5)$$

Where $c_{me}$ is the concentration in the mantle embryo, $c_{ce}$ is the concentration in the core of the embryo, and $c_b$ is its bulk composition. $F$ is the mass fraction of the body's core which is assumed to be 0.323 for the Earth. The effective partitioning of the species in embryos are assumed to stay constant and are defined as:

$$\frac{C_{ce}}{C_{me}} = D_c \quad (S6)$$

The mass conservation of an element, $c$, in the accreting Earth's core and mantle are integrated numerically following the approaches developed in[61]:

$$\frac{d}{dt}((1 - F)Mc_m) = [(1 - F)c_{me} + \varepsilon F(c_{ce} - D_c c_m)]\frac{dM}{dt} \quad (S7)$$

$$\frac{d}{dt}(FMc_c) = +[\varepsilon FD_c c_m + (1 - \varepsilon)Fc_{ce}]\frac{dM}{dT} \quad (S8)$$

Where $c_m$ and $c_c$ are the concentration of chemical species in the mantle and in the core of the Earth. For the growing Earth, $D_c$ evolves as a function of P, T, and other thermodynamic variables. Partial equilibration due to incomplete mixing is introduced into the model by the parameter $\varepsilon$, which defines the equilibrium efficiency between metal and silicate (i.e., mass exchange between metal and silicate normalized by its maximum possible value[61].

The effects of partial equilibration were quantified following the formalism developed from the results of fluid dynamics experiments[64]. Efficient equilibration of both metal and silicate requires that the large volumes of iron from impactor cores mix with molten silicates down to small scales. The mixing of metal and silicate is quantified through turbulent entrainment[64]. Following this work, a term of equilibration efficiency ($\varepsilon$) can be calculated at each step of accretion and core-mantle equilibration at the bottom of the magma ocean. This parameter is used in our modeling and equilibration of metal and silicate is accordingly dependent on (1) the siderophily of the considered element at P–T conditions of equilibration, (2) the size of the impactor, (3) the depth of the magma, (4) the entrainment coefficient taken after[64] (i.e., $\alpha = 0.25$). The $\varepsilon$ term is also calculated for the last giant impact considered in this work and as the impactor core diameter approaches the depth of the magma ocean, a low degree of re-equilibration is predicted. In contrast, impactors with small sizes relative to the depth of the magma ocean, re-equilibrate efficiently.

For a highly siderophile element, equilibration is efficient only if the metal mixes with a much larger mass of silicate (roughly D times its mass of silicate). The efficiency of metal–silicate equilibration has been shown to be strongly lowered when the size of the impactors approaches the thickness of the magma ocean[64]. Thus, only a fraction of the metal and silicate re-equilibrated at P–T conditions of the base of the magma ocean notably during the accretion from giant impacts. To evaluate the effects of partial equilibration on the final Pt content of the BSE, we considered a few different scenarios including having the Earth accreted mostly

from large impactors (results are discussed further below together with details of partial equilibration modeling and shown in Supplementary Fig. 4).

After each final core-mantle equilibration event in each model run, a complementary chondritic late accretion mass is also estimated in order to account for the present mantle's total platinum content[6]. The composition of late accretion is assumed to be similar to CI composition (e.g.,[65]) as supported by recent works arguing that carbonaceous chondrites best matches the compositions of highly volatile elements such as N, C, and H[66,67] and the overabundances of moderately volatile elements such as S, Se, and Te[68,69]. Os isotopic signatures have been interpreted to support both H and CI compositions[6,70] while Ru isotopic signatures have been shown to favor both a CI composition[40] and an inner solar system origin[71] for late accretion components. However, considering other types of chondritic materials (ordinary or enstatite chondrites) as possible sources for the late accretion would only marginally affect the estimates of late accretion required to account for the Pt content of the BSE as the chondrites contain fairly homogeneous Pt composition[62].

**Model results.** Full core-mantle equilibration ($\varepsilon = 1$) along an average geotherm can account for the Pt content of the mantle if core-mantle equilibration occurs between 54 to 59 GPa (Fig. 3a). If equilibration occurs at P of ~55 GPa, the required late accretion addition is ~0.38% $M_E$, coincident with the lower bound estimate of late accretion mass[2]. Above ~55 GPa, core-mantle equilibration leads to Pt content incompatible with the minimum amount of late accretion required to explain abundances of other HSE. The blue horizontal line in Fig. 3a shows the late accretion mass estimated with the chondritic relative abundances of other HSE[5].

In Fig. 3b, the white and red circles correspond to the minimum amount of Pt left in the mantle from core formation models that account for a suite of moderately and slightly siderophile elements (Ni, Co, V, and Cr) that have been extensively used in previous works to constrain the P, T, and fO2 conditions of Earth's core formation (e.g., 21, 29, 30). All results from the modeling for any P–T–fO2 conditions fall on this black line. For instance, if accretion started at ΔIW-4.5 along a cool geotherm, a minimum amount of Pt (0.8 ppb) can be obtained for a final pressure of core-mantle equilibration at 48 GPa compatible with the observed contents of Ni–Co–V–Cr of the BSE. But this solution gives an Earth's core containing low amounts of O (0.5 wt.%) and Si (3.9 wt.%) incompatible with seismology[32]. Models leading to acceptable solution for both the budget of siderophile elements (Ni, Co, V, and Cr) in the mantle and light elements (O, Si) in the core all lead to minimum values incompatible with the late accretion mass required to account for other HSE and Os isotopes. This shows that additional mechanisms (e.g., disproportionation) were necessary in early magma oceans to remove the excess of Pt leftover from core formation.

Partial equilibration between metal cores of impactors in the magma ocean could lower the amount of platinum which accumulated in the mantle at a given P–T condition. During partial equilibration, metal cores of impactors do not fully disperse, and large chunks of metal are sequestered to the core without chemically interacting with the magma ocean. This occurs during the accretion of large impactors which is generally considered to have occurred towards the later stages of Earth's accretion (e.g.,[8]). Using partial equilibration formalism from[64], core formation could lead to 2–3 ppb Pt between 53 and 59 GPa and could account for the Pt content of the BSE at higher pressures, between 75 and 80 GPa (Supplementary Fig. 4). Thus, in principle partial equilibration could lead to Pt content compatible with late accretion addition if equilibration occurred between 53 and 59 GPa. However, in order to account for moderately siderophile elements, the final pressure of core-mantle equilibration has to be shifted to higher pressures, between 75 and 85 GPa. Accordingly, partial equilibration cannot provide a self-consistent mechanism to lower the amount of Pt in the BSE due to core-mantle equilibration. The models always result in an overabundance of Pt within the P–T–fO2 solution space that can also account for other moderately siderophile elements. Above 55 GPa (full equilibration, Fig. 3a) or above around 68 GPa (partial equilibration, Supplementary Fig. 4) there is too much Pt and not enough late accreting components to account for all HSE.

Error envelopes for platinum concentrations presented with the model results (Fig. 3a, b and Supplementary Fig. 4) are based on the uncertainties from the regression analysis for platinum (see Eq. 2).

**Pt Removal due to disproportionation of $Fe^{2+}$.** The mantle's Pt content is calculated after segregation by different amounts of precipitated Fe (resulting from the disproportionation of $Fe^{2+}$ to $Fe^{3+}$ and $Fe^0$) using a single-stage mass balance calculation based on Eq. (S5)[36]. In this calculation $c_b$ is the platinum composition of the mantle after core formation and $c_{me}$ is the final concentration of the mantle after interaction with a fraction $F$ of metallic iron. The partition coefficient is equivalent to $c_{ce}/c_{me}$ and values used are based on the partitioning expression (main text Eq. 2). The lack of HSE and other impurities in the precipitated Fe allow this process to be efficient at stripping out Pt even in a shallow magma ocean. This is in contrast to the cores of impacting planetesimals which already contain a full complement of HSE. In the case of inefficient mixing and equilibration of Fe droplets in the magma ocean more Fe precipitation would be required to remove HSEs. For example, if mixing is only 50% efficient that would roughly double the amount of Fe required to remove the Pt accumulated in both deep or shallow magma ocean conditions (~1 wt.% Fe for 30 ppb of Pt in a deep magma ocean).

*The evolution of Pt/Os and $^{186}Os/^{188}Os$ of the mantle.* The expected Pt/Os ratio of the pre-late accretion mantle was determined using the partition behavior for Pt determined in this study (main text Eq. 2) and the partitioning expression for Os from earlier work[31] as a single-stage core formation calculation (based on Eq. S5). The P–T of core-mantle equilbration considered are $P = 60$ GPa and $T = 3500$ K. Initial bulk Earth concentrations of Pt and Os are based on a chondritic model of the Earth[62]. $^{190}Pt$ decays to $^{186}Os$ with a very long half life ($6.5 \times 10^{11}$ years) and is considered stable on timescales of the Earth's formation[39]. The Pt/Os ratios after core formation range from ~60 to 8.6 which would result in more radiogenic $^{186}Os/^{188}Os$ than the BSE over time. The BSE $^{186}Os/^{188}Os$ is similar to chondrites but samples derived from some plume lavas show suprachondritic values (Fig. 4b). We calculated the average $^{186}Os/^{188}Os$ values that would result from mixing between the modern upper mantle and varying fractions (10 to 30%) of a pre-late veneer mantle reservoir that did not experience disproportionation.

## Data availability

The datasets generated during and/or analysed during the current study are available from the authors on reasonable request.

## Code availability

Modeling was conducted using commercial software MATLAB.

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

## Acknowledgements
This work was supported by French state funds managed by ANR within the Investissements d'Avenir program under reference ANR-11-IDEX-0004-02, and within the framework of the Cluster of Excellence MATISSE led by Sorbonne Universités. J.S. acknowledges support from the French National Research Agency (ANR project Vol-Terre, grant no. ANR-14-CE33-0017-01). The authors are grateful for the use of the National NanoSIMS facility at the MNHN, established by funds from the CNRS, Region Ile de France, Ministère délégué à l'Enseignement supérieur et à la Recherche. The authors acknowledge Carol La, Jean-Pierre Lorand, Adriana Gonzalez-Cano, and Sylvain Pont for assistance with standards characterization and measurements. T-A.S. acknowledges discussions with Neil Bennett and Rebecca A. Fischer.

## Author contributions
J.S., L.R., and G.F. jointly supervised the work. T-A.S., J.S., L.R., S.B., B.D., and G.F. contributed to sample preparation and characterization. T-A.S., J.S., and J.M.D.D. carried out modeling and wrote the paper.

## Competing interests
The authors declare no competing interests.
