## [Peer Review File · Nature Communications]

REVIEWER COMMENTS

Reviewer #1 (Remarks to the Author):

The manuscript by Suer et al. reports a metal-silicate partitioning coefficient of Pt through wide pressure range which covers the conditions of Earth's core formation. The results are applied to core formation models and a more amount of Pt than the present mantle abundance is left in the mantle. To reconcile the excess Pt amount, iron segregation after the core formation is suggested. It is implied that an anomaly in Os isotope might be produced by the heterogeneity of the mantle due to the iron segregation.

Metal-silicate partitioning experiments in DAC are already well established and this research is similarly performed. Results of DAC experiments show typical textures of similar experiments, which supports the credibility of the experiments. Chemical characterization is carefully done and technically interesting. Measurements of highly siderophile elements abundance in silicates have been challenging for DAC experiments, but use of FIB and NanoSIMS in this study enables the data to be reliable even in a small and low concentration sample. While more careful interpretation may be required, an attempt to characterize metallic particles in silicates by TEM are also interesting because those particles were often found to be problematic in previous studies.

My concern is the impact of geophysical implication. The core formation process is important for the Earth science community and the manuscript can potentially contribute to it. Behavior of highly siderophile elements is a critical factor to assess the late veneer/accretion story. However, the present manuscript just supports the existing theories which are already expected by previous studies. Also, the wide possible range of core formation conditions makes it difficult to obtain robust conclusion about the core formation process.

In my understanding, final FeO concentration in the mantle can be obtained through the core formation modeling. There should be excess FeO in the mantle after the core formation compared to the present value if the iron segregation occurred afterwards. The amount of excess FeO in each core formation condition and the amount of segregated Fe required for producing the present Pt amount (like Fig 4A) must be similar, which may be able to constrain the core formation condition. If some constraints like this can be obtained by considering the core formation and iron segregation together, it will be highly valuable.

Here are other specific comments.

About experiments:

1. In line 115-116, authors write "The high average Fe/Pt ratios of the particles (~ 12.5) also suggest that they have negligible influence on the platinum concentration in the silicate.". Even though Fe/Pt is high, Pt concentration in metal particles is still higher than that in silicates. Was it just ignored during NanoSIMS measurement, or taken into account by averaging?
2. In line 527-528 and Fig S1, final temperature is kept for only ~ 10 sec although it is superliquidus for a longer period. Chemical equilibrium should be assessed by the duration of final temperature.
3. About the temperature measurement written in line 523-524, authors should provide more detailed information because the temperature dependence is an important parameter in the manuscript. Temperature gradient is often a problem for LH-DAC. If "highest temperature" means the center of laser heated spot, the temperature at the metal-silicate interface could be lower than reported temperature.

4. About NanoSIMS measurements, the sample is just placed on a silicon wafer in Fig S2. How is the horizontality of a sample surface assured? Even though it was polished by FIB, it could be tilted when it is attached.

5. In line 107-108 and 627-629, how were ROIs chosen? It must be considered carefully because metallic particles could affect the measurement.

About parameterization:

6. In line 142, authors write "the effect of pressure on its partitioning behavior is negligible". However, high temperature results in the present study are obtained at higher pressure than the previous study. Isothermal experiments should be performed to separate the effect of pressure and temperature.

7. Simple parameterization of D_{Pt} is adopted in the manuscript. Effects of light elements should be treated carefully because the effect of sulfur was reported previously (Laurenz et al. 2016).

About the core formation modeling

8. In line 735-737, a final impactor is assumed to be larger than the other steps. Equilibrium efficiency is often set to low value for such a large impactor. Authors should discuss how it is treated in both full equilibration and partial equilibration models.

9. In a case of shallow magma ocean in iron segregation, most of the mantle is already solidified. How do authors treat Pt in the solid mantle? Is it assumed that all Pt is contained in the magma ocean?

Minor points:

10. In Eq.S7, F is probably lacking in the second term of the right side. Also, one of brackets in Eq.S8 must be wrong.

11. In line 163, "0.34ME" means 0.34%ME?

Reviewer #2 (Remarks to the Author):

Review of Nature Communications ms. 274927, "Reconciling metal-silicate partitioning and late accretion in the Earth" by Suer et al.

This paper presents new state-of-the-art partitioning data for Pt at lower mantle P and T conditions pertinent to core formation in the Earth. The results show that under these conditions, Pt is much less siderophile than it is at near surface temperatures and pressures. Thus this change in D value with depth could at first glance explain the higher than expected mantle abundances of Pt. However, as the authors point out, it is not just Pt, but all of the highly siderophile elements (HSE) that are enriched relative to what would be expected from shallow partitioning. Moreover these elements are present in chondritic relative proportions, an observation that has long been taken as evidence for a small amount of late accretion of chondritic material after the completion of core formation. So there is a major contradiction in the available data that needs explanation. The authors suggest that this contradiction could be resolved by sulfide exsolution and/or Fe disproportionation reactions in a magma ocean that would create Pt-sequestering phases that could then be incorporated into the core.

The paper is well-written, enjoyable to read, and in most respects clear (but see below). It addresses a major issue related to Earth formation processes. The contradiction brought to light by these new data has been hinted at by earlier Pt partitioning data at shallow mantle conditions, but this is the first

data that I know of that directly demonstrate decreased Pt partition coefficients at ultra high P and T. For all of these reasons, I think this paper merits publication in Nature Communications, after some clarifications.

I reviewed an earlier version of this manuscript for another journal. In this new version, the authors have responded to many of my concerns. Overall the paper is easier to follow, and notably, the supplemental material now includes more detailed information about the modeling.

Nevertheless there is one significant issue that was not adequately addressed. Figure 3, which is a key diagram, remains very difficult to understand. This is quite frustrating, as the diagram is apparently simple. My guess (but it's only a guess) is that to get the solid black curve in Fig. 3a, at each pressure the corresponding temperature was determined from the geotherm (but which one? Three possibilities are given in Eq. S2 - S4.) and the D value for Pt was then obtained from Eq. 2. Then I guess a single stage silicate-metal equilibration step was performed using the mass balance equation (S5), assuming a BSE value of 7.6 ppb and the current mass fraction of the Earth's core. Is this indeed what was done? If so, this should be explained explicitly, the reader should not be obliged to guess. If a more complex multi-stage equilibration model was assumed, this should be described clearly.

Interpretation of the blue curve is even more challenging. My understanding, based on line 426 in the caption to Fig. 3, is that the blue curve is the amount of late accreted material that would have to be added to bring the total amount of Pt up to the measured BSE Pt content. But at 60 GPa, for example, core formation would be expected to leave a mantle with 10 ppb Pt, the upper limit of the BSE field, so no additional material would have to be added. However the blue curve shows addition of about 0.34% material at that pressure. In fact, I'm not sure that the blue curve is very helpful (in any case it should not be placed in the same graph as the black curve as this adds unnecessary confusion). I think that what the authors are trying to show is that deep metal-silicate partitioning followed by late accretion would result in an excess of Pt relative to the other HSE. A simpler way of doing this might be to show the Pt/Os (or Pt/Ir) ratio that would be obtained as a function of pressure by adding the Pt left over from core formation to the material added during late accretion, and comparing this with the BSE Pt/Os ratio.

Fig. 3b poses its own challenges. I presume the black solid line is just based on the D value obtained from Eq. 2, converted to Pt abundances by assuming single stage equilibration between silicate and metal. But in that case, why do all of the very different models plot on this line? Is a complex accretion history assumed for each case, followed by total equilibration at the final temperature? Again, more explanation is needed.

I have added many specific comments on the attached annotated pdf file, on both the main text and the supplementary material. For example (lines 62-64), I don't understand why positive 182W anomalies should be associated with high HSE contents (the opposite might be expected as Hf/W ratios are elevated by W sequestration in the core). It also seems odd (lines 827-828) that partial equilibration with impactor cores would lead to lower Pt contents in the mantle than is expected after full equilibration. Here and elsewhere I may just be misunderstanding what the authors mean to say, but I suggest that they look through these annotated comments carefully because they indicate where the manuscript is not clear.

Laurie Reisberg

Reviewer #3 (Remarks to the Author):

The article is based on an experimental investigation of the metal/silicates partitioning behaviour of

platinum (Pt) at high pressure and temperature.

Platinum is found to be significantly less siderophile at high temperatures. Using the experimentally determined Pt metal-silicate partitioning coefficients, it is found that core formation models consistent with geochemical and geophysical constraints predict an excess of Pt in the mantle. This is at odds with the chondritic relative abundances of Highly Siderophile Elements (HSE), which requires that the final Pt mantle content has been set by a late addition of chondritic material (late veneer).

The Pt sequestered in the Mantle during core formation must therefore have been removed by some additional mechanism, which the authors argue could be iron disproportion or sulfide segregation. The authors also argue that the suprachondritic $^{186}/^{188}$ Os isotopes ratio of Hawaiian magmas could be explained if a Pt-enriched reservoir that did not experience disproportionation could have been preserved.

This is a very nice and interesting paper, with clearly important results, and I believe it is well worth publishing in Nature Communications. I have only relatively minor comments and questions.

> I have several questions and comments concerning the mechanism of Pt removal due to Fe disproportion:

- The estimation of Pt removal due to Fe disproportion seems a bit optimistic to me since the calculation (explained in section 5c of the Methods) assumes that the precipitated Fe droplets equilibrate with the full mantle. If disproportion happens in a magma ocean, then the solid part of the mantle is unlikely to equilibrate with the precipitated iron, and would probably keep its Pt. If instead disproportion happens in the solid part of the mantle, then it is far from obvious that the metal could efficiently segregate from the silicates and reach the core.

I would thus think that disproportion would at most remove the excess Pt from the magma ocean.

If correct, this would significantly affect the amount of Pt removal calculated and presented in figure 4. The Deep Magma Ocean case ($P \sim 65$ GPa) corresponds to a magma ocean of roughly 70% of the total mass of the mantle; if the solid part of the mantle keeps its Pt, then the final Pt concentration should be no less than 30% of its pre-removal concentration. In the Shallow Magma Ocean case, the magma ocean amounts to roughly 35% of the total mass of the mantle, and the final Pt concentration should be no less than 65% of its pre-removal concentration.

There might be an optimal magma ocean depth: in a shallow magma ocean, the Pt metal/silicate partitioning coefficient is high, but the mass of equilibrated mantle is low; in a deeper magma ocean, the Pt metal/silicate partitioning coefficient is lower, but the mass of equilibrated mantle is higher.

This effect could easily be included in the calculations leading to figure 4. I therefore believe that the authors should either estimate the amount of Pt removal by taking into account partial equilibration of the mantle, or give arguments in support of full mantle equilibration.

- Shouldn't Fe disproportion also affect the abundance of moderately siderophile elements? The precipitated iron should also carry some of these elements to the core. My guess is that the effect would be small since the metal-silicates partitioning coefficients of these elements isn't very high, but maybe this should be quantified.

- In the model, Fe disproportion is assumed to happen after core formation. However, Fe disproportion could start as soon as the pressure in Earth's mantle is high enough, and it should probably be seen as concomitant of core formation.

Would this affect the efficiency of Pt removal ?

> The authors mention sulfide segregation as a possible way of removing the excess Pt, but this is not quantified in the text. Would this mechanism be as efficient as iron exsolution ?

REVIEWER COMMENTS

Reviewer #1 (Remarks to the Author):

The manuscript by Suer et al. reports a metal-silicate partitioning coefficient of Pt through wide pressure range which covers the conditions of Earth's core formation. The results are applied to core formation models and a more amount of Pt than the present mantle abundance is left in the mantle. To reconcile the excess Pt amount, iron segregation after the core formation is suggested. It is implied that an anomaly in Os isotope might be produced by the heterogeneity of the mantle due to the iron segregation.

Metal-silicate partitioning experiments in DAC are already well established and this research is similarly performed. Results of DAC experiments show typical textures of similar experiments, which supports the credibility of the experiments. Chemical characterization is carefully done and technically interesting. Measurements of highly siderophile elements abundance in silicates have been challenging for DAC experiments, but use of FIB and NanoSIMS in this study enables the data to be reliable even in a small and low concentration sample. While more careful interpretation may be required, an attempt to characterize metallic particles in silicates by TEM are also interesting because those particles were often found to be problematic in previous studies.

My concern is the impact of geophysical implication. The core formation process is important for the Earth science community and the manuscript can potentially contribute to it. Behavior of highly siderophile elements is a critical factor to assess the late veneer/accretion story. However, the present manuscript just supports the existing theories which are already expected by previous studies. Also, the wide possible range of core formation conditions makes it difficult to obtain robust conclusion about the core formation process.

In my understanding, final FeO concentration in the mantle can be obtained through the core formation modeling. There should be excess FeO in the mantle after the core formation compared to the present value if the iron segregation occurred afterwards. The amount of excess FeO in each core formation condition and the amount of segregated Fe required for producing the present Pt amount (like Fig 4A) must be similar, which may be able to constrain the core formation condition. If some constraints like this can be obtained by considering the core formation and iron segregation together, it will be highly valuable.

Thanks for raising these questions. **These are very interesting points which have now been clarified in the main manuscript and the Methods and Supporting Materials (SOM). A paragraph was added to the discussion (lines 212 – 221) regarding those concerns and more details added to SOM.** However, we have already explored the conditions of Earth's core formation constrained from the present work. Using the present Pt partitioning dataset combined with previous partitioning datasets for other siderophile elements and mineral physics constraints allowed us to further refine the conditions of Earth's core formation. We propose the following conclusions which we believe are being raised for the first time using the partitioning data of a highly siderophile element (HSE):

(1) The Pt content of the present bulk silicate Earth (BSE) can be obtained at P–T conditions of core-mantle equilibration that can also account for other moderately siderophile elements such as Ni and Co (45–65 GPa at liquidus T; and at higher P–T conditions if partial equilibration between metal and silicate is considered) without late accretion (Figure 3a).

(2) The Os isotopes systematics and the chondritic relative abundances of other HSEs require the addition of late accreting components. When considering Ni–Co constraints on the conditions of Earth’s core formation this allows for a maximum late accreting addition of ~ 0.4 wt.% of Earth’s mass (Figure 3a).

(3) All models accounting for the P–T–fO₂ conditions of Earth’s core formation imposed by other geochemical (i.e., the abundances of slightly siderophile elements such as Cr and V) and geophysical constraints (i.e., the density and sound velocities of the Earth’s core as a function of its light elements composition) lead to an excess of Pt in the Earth’s mantle even before addition of the late veneer (Figure 3b). This strongly suggests that self-reducing processes (i.e., iron disproportionation/sulfide saturation) were efficient after core formation in order to remove the excess Pt, explain the modern mantle HSEs signature, and possibly raise the redox state of the mantle from ~IW–2 (after core formation) to its present-day value (~ QFM).

We believe that these are a unique set of interesting conclusions that already provide constraints on the conditions of early Earth differentiation. Regarding the predicted range of P–T–fO₂–X conditions that can produce the present pattern of siderophile elements in the BSE (i.e., Ni, Co, V, Cr), the present partitioning data cannot be used to place further constraints on the conditions of Earth’s core formation.

To more specifically address Reviewer’s 1 comment, it would be valuable if the amount of Fe required to segregate HSEs could be used to place additional constraints on the final redox state of core formation and mantle redox state evolution. We now elaborate more on this application of our results in the discussion (**lines 212 – 221**). However, we show that a small amount of mantle FeO from the BSE precipitated to metal Fe (i.e., ~ 0.5 wt.%) is sufficient to remove the excess Pt produced in all core formation scenarios. Such limited amount of precipitated iron is unlikely to significantly modify the conditions of Earth’s core formation and notably the final fO₂ of core-mantle equilibrium imposed by the present amount of FeO in the BSE (i.e., 8 wt.% FeO). Moreover, Pt partitioning is not dependent of the redox state of the system (i.e., zero valence state for Pt is predicted in the silicate melt) and considering a slightly different amount of FeO in the mantle after core formation (and before excess Pt removal) would not affect the predicted Pt excess of the BSE from this work. Armstrong et al. (2019), showed that up to 3 wt. % or more of segregated Fe in a deep magma ocean would allow the mantle’s Fe³⁺/Total Fe ratio to match that of the modern-day mantle. Partitioning coefficients of other HSEs (especially the less siderophile HSEs) would have to be studied at similar conditions to the ones in our study in order to further constrain the amount of Fe required to remove all of the HSEs. For example, the amount of Pd expected after core formation is up to 100 ppb based on measurements carried out on LHDAC samples (Suer et al., in prep). This amount of Pd could be also stripped out by ≤ 1 wt. % of Fe precipitation.

Here are other specific comments.

About experiments:

1. In line 115–116, authors write “The high average Fe/Pt ratios of the particles (~12.5) also

suggest that they have negligible influence on the platinum concentration in the silicate.”. Even though Fe/Pt is high, Pt concentration in metal particles is still higher than that in silicates. Was it just ignored during NanoSIMS measurement, or taken into account by averaging?

The Pt amount in the nanoparticles is higher than in the silicate. However, the Fe/Pt ratio of these inclusions is high (i.e., very far from pure Pt inclusions) and at the extreme P–T conditions of our DAC experiments the amount of Pt in the silicate melt is quite high (i.e., ~1000s ppm) compared to the very low Pt content usually observed in lower P–T experiments (few ppm). As the volume represented by these inclusions of quench origin is very small with respect to the volume of silicate melt for a NanoSIMS measurement, we expect that the presence of these inclusions would have a negligible effect on the measured Pt contents. This is confirmed as Pt content of silicate regions without obvious nanoparticles and with nanoparticles did not show noticeable differences. The Pt content of nanoparticles of quench origin are averaged into the overall platinum concentration of the silicate.

2. In line 527-528 and Fig S1, final temperature is kept for only ~10 sec although it is superliquidus for a longer period. Chemical equilibrium should be assessed by the duration of final temperature.

The extreme temperatures of LHDAC experiments at superliquidus conditions strongly favor chemical diffusion. For instance, Nomura et al. (2011) estimated a diffusion length of ~ 40 μm in one second in silicate melt for Fe. The radii of the melt pockets in our experiments are around 15–20 μm . It has been shown in numerous previous LHDAC works (e.g., Nomura et al., 2011; Bouhifd et al., 2011; Siebert et al., 2012; 2013; Fischer et al., 2015; Chidester et al., 2017; Blanchard et al., 2017) that chemical equilibration was reached in few seconds. This is well confirmed by the chemical homogeneity of metal and silicate phases in run products of these experiments (where final T was kept for more than 10 s in all experiments). **Duration and chemical equilibration of experiments are now further discussed in the SOM (Lines 601-607).**

3. About the temperature measurement written in line 523-524, authors should provide more detailed information because the temperature dependence is an important parameter in the manuscript. Temperature gradient is often a problem for LH-DAC. If “highest temperature” means the center of laser heated spot, the temperature at the metal-silicate interface could be lower than reported temperature.

Temperatures were measured with the spectro-radiometric analysis of thermal radiation spectra in the spectral range 450-850 nm. Measured temperatures were the average of the central 5 microns of the hot spot. With emissivity assumed to be independent of wavelength in the Planck radiation function, temperature uncertainties are of the order of ± 200 K. Temperature gradients are present during LHDAC experiments. We did not have the ability (4-color imaging system) to quantify this gradient but these variations are instead considered in the uncertainties for the temperature measurements which incorporate the difference between the measured temperatures on the two sides of the sample, an analytical uncertainty of 100 K (Shen et al., 2001), and an uncertainty associated with the correction for axial temperature gradients (Campbell et al., 2009).

The samples consist of a thin foil of (Fe, Pt) alloy sandwiched between two disks of silicate glass in controlled geometry. Temperature gradients are strong across a very thin boundary layer

between partially molten sample and the cold surrounding glassy material, which acts as chemical and thermal insulation. Axial temperature gradients cannot be avoided within the heated area but are strongly minimized due to the superliquidus conditions of experiments (e.g., Siebert et al., 2012; Fischer et al., 2015; Blanchard et al., 2017). **All these details are now discussed in the SOM as suggested by the reviewer (SOM Lines 582 - 594).**

4. About NanoSIMS measurements, the sample is just placed on a silicon wafer in Fig S2. How is the horizontality of a sample surface assured? Even though it was polished by FIB, it could be tilted when it is attached.

The sections were ion polished on both sides and were of equal thickness across the whole section before being deposited flat on the wafer and attached with an electron curing glue. Thus, every precaution was taken to ensure the flatness of the sample and this was evidenced by further SEM observations before NanoSIMS measurements. **This is now mentioned in the SOM (Line 626-630).**

5. In line 107-108 and 627-629, how were ROIs chosen? It must be considered carefully because metallic particles could affect the measurement.

ROIs used to derive the average platinum concentration were homogeneous in appearance and free of large metallic inclusions. In addition, NanoSIMS is a surface measurement technique so contamination from underlying substructures are avoided.

About parameterization:

6. In line 142, authors write “the effect of pressure on its partitioning behavior is negligible”. However, high temperature results in the present study are obtained at higher pressure than the previous study. Isothermal experiments should be performed to separate the effect of pressure and temperature.

Isothermal experiments at various pressures are difficult to achieve in LHDAC experiments. Above the liquidus of both metal and silicate there are strong temperature fluctuations (i.e., absorption jump of the sample and high emissivity of the heated region) and it is difficult to control the laser heating power in order to achieve a target temperature. However, there are four DAC experiments carried out at temperatures between 3900–4100 K and pressures between 43–85 GPa that can be regarded, to first order, as isothermal experiments. Figure 2 shows that the Pt partition coefficients of these experiments are strongly temperature dependent and that pressure effect is negligible. The multiple variable regression models allow us to consider the effects of all the variables simultaneously and separate the effects iteratively using a large dataset which includes the previous works and the extreme P–T experiments from this work.

7. Simple parameterization of DPt is adopted in the manuscript. Effects of light elements should be treated carefully because the effect of sulfur was reported previously (Laurenz et al. 2016).

At conditions of our experiments (i.e., high temperatures), there are significant amounts of light elements (i.e., O, Si, and S) in the metal. Through the parameterization (SOM Section 5), we modeled the potential effects of metallic composition (presence of light elements) on the Pt

partitioning. However, as reported in the manuscript, the effects of O, Si and S are found to be statistically negligible on Pt partitioning. The simpler temperature parameterization is due to elimination of the insignificant variables in the multi steps regression analysis. Laurenz et al., (2016) reported a resolvable negative effect of sulfur on Pt partitioning. This effect was inferred from experiments containing large amounts of S in the (Fe, S) alloys (i.e., S contents above the eutectic composition in the Fe–S system and even stoichiometric FeS liquids). At conditions of our experiments, the sulfur contents of metal are relatively small (~ 0.1 for four experiments and two experiments where the sulfur contents are very low below 0.03 (i.e., less than 2 wt.%)). Laurenz et al., (2016) does not predict a significant effect of sulfur on Pt partitioning, particularly at low sulfur compositions in our DAC data. **A short discussion regarding light elements effects on Pt partitioning is now given in the SOM (lines 800 – 810).**

About the core formation modeling

8. In line 735-737, a final impactor is assumed to be larger than the other steps. Equilibrium efficiency is often set to low value for such a large impactor. Authors should discuss how it is treated in both full equilibration and partial equilibration models.

This aspect of the modeling was already discussed in the supplementary materials, but **we have now provided more details on the effect of impactor sizes on equilibration efficiency and how they affect the model results (lines 872 – 884 and 929 – 933)**. We quantified the effects of partial equilibration following the formalism developed in Deguen et al., (2014). Efficient equilibration of both metal and silicate requires that the large volumes of iron from impactor cores mix with molten silicates down at low scales. Using fluid dynamics experiments, Deguen et al., (2014) quantifies the mixing of metal and silicate through turbulent entrainment. Following this work, a term of equilibration efficiency (ϵ) can be calculated at each step of accretion and core-mantle equilibration at the bottom of the magma ocean. This parameter is used in our modeling and equilibration of metal and silicate is accordingly dependent on: (1) the siderophility of the considered element at P–T conditions of equilibration, (2) the size of the impactor, (3) the depth of the magma, (4) the entrainment coefficient taken after Deguen et al., 2014 (i.e., $\alpha = 0.25$). The ϵ term is also calculated for the last giant impact considered in this work and as the impactor core diameter approaches the depth of the magma ocean, a low degree of re-equilibration is predicted. In contrast, impactors with small sizes relative to the depth of the magma ocean, re-equilibrate efficiently.

9. In a case of shallow magma ocean in iron segregation, most of the mantle is already solidified. How do authors treat Pt in the solid mantle? Is it assumed that all Pt is contained in the magma ocean?

As with most previous works using models of continuous core formation during accretion (e.g., Wade and Wood, 2005; Corgne et al., 2008; Tuff et al., 2011; Siebert et al., 2012; 2013; Mahan et al., 2018), we assumed a depth of roughly 40 % of the depth of the core-mantle boundary (CMB) for the magma ocean. The full mantle is involved in each equilibration step in this type of model and Pt is accordingly contained in the bulk BSE.

Minor points:

10. In Eq.S7, F is probably lacking in the second term of the right side. Also, one of brackets in

Eq.S8 must be wrong.

Thanks for noticing this typo. This has now been corrected in the supplement.

11. In line 163, “0.34ME” means 0.34%ME?

This is correct. This has now been corrected in the manuscript.

Reviewer #2 (Remarks to the Author):

Review of Nature Communications ms. 274927, "Reconciling metal-silicate partitioning and late accretion in the Earth" by Suer et al.

This paper presents new state-of-the-art partitioning data for Pt at lower mantle P and T conditions pertinent to core formation in the Earth. The results show that under these conditions, Pt is much less siderophile than it is at near surface temperatures and pressures. Thus, this change in D value with depth could at first glance explain the higher than expected mantle abundances of Pt. However, as the authors point out, it is not just Pt, but all of the highly siderophile elements (HSE) that are enriched relative to what would be expected from shallow partitioning. Moreover, these elements are present in chondritic relative proportions, an observation that has long been taken as evidence for a small amount of late accretion of chondritic material after the completion of core formation. So, there is a major contradiction in the available data that needs explanation. The authors suggest that this contradiction could be resolved by sulfide exsolution and/or Fe disproportionation reactions in a magma ocean that would create Pt-sequestering phases that could then be incorporated into the core.

The paper is well-written, enjoyable to read, and in most respects clear (but see below). It addresses a major issue related to Earth formation processes. The contradiction brought to light by these new data has been hinted at by earlier Pt partitioning data at shallow mantle conditions, but this is the first data that I know of that directly demonstrate decreased Pt partition coefficients at ultra-high P and T. For all of these reasons, I think this paper merits publication in Nature Communications, after some clarifications.

I reviewed an earlier version of this manuscript for another journal. In this new version, the authors have responded to many of my concerns. Overall the paper is easier to follow, and notably, the supplemental material now includes more detailed information about the modeling.

Nevertheless there is one significant issue that was not adequately addressed. Figure 3, which is a key diagram, remains very difficult to understand. This is quite frustrating, as the diagram is apparently simple.

This is indeed a very important point that needs to be improved in the new version of the manuscript. **We have now added more explanations in the manuscript and in the SOM to address the concerns of Reviewers #2 and to give readers the information required to correctly interpret Figure 3.**

-My guess (but it's only a guess) is that to get the solid black curve in Fig. 3a, at each pressure the corresponding temperature was determined from the geotherm (but which one? Three possibilities are given in Eq. S2 - S4.)

In Figure 3a, each pressure corresponds to the last pressure of equilibration for a full continuous core formation history (See SOM lines 816–828). In this figure, the temperature follows the liquidus temperature of pyrolite type compound averaged from the experimental studies of Fiquet et al., (2010) and Andrault et al., (2011). This approach follows the same method developed in numerous previous publications (e.g., Siebert et al., 2012; 2013; 2018; Badro et al., 2015; Blanchard et al., 2017; Mahan et al., 2018). **We have now mentioned which temperature geotherm was considered in the caption of Figure 3a (Lines 511–515).**

-and the D value for Pt was then obtained from Eq. 2. Then I guess a single stage silicate-metal equilibration step was performed using the mass balance equation (S5), assuming a BSE value of 7.6 ppb and the current mass fraction of the Earth's core. Is this indeed what was done? If so, this should be explained explicitly, the reader should not be obliged to guess. If a more complex multi-stage equilibration model was assumed, this should be described clearly.

A D value for Pt which depends on P–T conditions at the base of the magma ocean is applied at each incremental step of accretion and core formation. The concentration of Pt in both the BSE and Earth's core are then integrated over the full accretion and core formation history, computed using equations S7 and S8. A final D for Pt can be obtained as a function of last pressure of equilibration, producing a continuous core formation model. The details of the modeling are now fully given and detailed in the supplements (i.e., Part 5a of the supplements: Core formation and accretion modeling) and readers are also referred to previous publications using strictly similar modeling of elemental partitioning during accretion and core formation (e.g., Siebert et al., 2013; Badro et al., 2015; Suer et al., 2017; Siebert et al., 2018). Space limitations does not allow for a more detailed description in the figure caption and main manuscript.

Interpretation of the blue curve is even more challenging. My understanding, based on line 426 in the caption to Fig. 3, is that the blue curve is the amount of late accreted material that would have to be added to bring the total amount of Pt up to the measured BSE Pt content. But at 60 GPa, for example, core formation would be expected to leave a mantle with 10 ppb Pt, the upper limit of the BSE field, so no additional material would have to be added. However, the blue curve shows addition of about 0.34% material at that pressure. In fact, I'm not sure that the blue curve is very helpful (in any case it should not be placed in the same graph as the black curve as this adds unnecessary confusion).

- This is a very relevant comment and this needed to be clarified for readers. The blue line represents the maximum amount of late veneer that can be added after core formation is finished. At each pressure, the blue curve corresponds to the maximum late accretion material that can complete the Pt leftover from core formation and not exceed the observed amount of Pt measured in the present mantle. The lower bound of Pt leftover from core formation that is considered for the late accretion mass, this is why at 60 GPa it is still possible to add some late accreting component as. **This was missing and is now mentioned explicitly in the caption of Figure 3a.**

We think that the blue line is still an important feature to be presented in the main MS. It illustrated an important conclusion of our work: (1) If available constraints from other siderophile elements (i.e. moderately siderophile Ni and Co or slightly siderophile V and Cr) and observables from seismological models (i.e. density and sound velocities) are not considered, the Pt content of the present mantle can be explained by a hybrid model by which the Pt and HSE abundances in the BSE are a combined result of both metal-silicate partitioning as well as an overprint by late accretion. This supports the conclusions of some previous works conducted at lower P-T conditions (Richter et al., 2008; Brenan and McDonough, 2009; Mann et al., 2012; Bennett et al., 2014) (2) However, we here also show that above ~55 GPa, core-mantle equilibration leads to Pt content incompatible with the minimum mass of late accretion required to explain abundances of other HSE. **These supplementary explanations are now provided both in the MS (Lines 166–172) and in the SOM (Lines 910 – 914). A section on late accretion has been added in the modeling section 5a, Lines 894 – 904).**

I think that what the authors are trying to show is that deep metal-silicate partitioning followed by late accretion would result in an excess of Pt relative to the other HSE. A simpler way of doing this might be to show the Pt/Os (or Pt/Ir) ratio that would be obtained as a function of pressure by adding the Pt left over from core formation to the material added during late accretion and comparing this with the BSE Pt/Os ratio.

Thanks for this comment. We show the Pt/Os ratio that would result from mixing material from core formation and late accretion in Figure 4a. The uncertainties are large due to lack of high P–T data for Os or Ir. Following from this, we did not think that having Pt/Os or Pt/Ir this on a separate figure would be very useful. However, this would be interesting for future follow up work if partition coefficients for Os and Ir were obtained at high pressures and temperatures in LHDAC experiments.

Fig. 3b poses its own challenges. I presume the black solid line is just based on the D value obtained from Eq. 2, converted to Pt abundances by assuming single stage equilibration between silicate and metal. But in that case, why do all of the very different models plot on this line? Is a complex accretion history assumed for each case, followed by total equilibration at the final temperature? Again, more explanation is needed.

We have now elaborated more in the captions and SOM on the core formation model in order to make it clearer for readers to understand. The D's from Equation 2 are integrated with Equations S7 and 8 over the growth of the Earth. As discussed above, in a previous question raised by Reviewer #2, the final Pt content is not obtained from single stage core-mantle equilibration. The final Pt content of the BSE is calculated from a more realistic view of core formation. It represents the integrated amount of Pt obtained for a complete continuous core formation history during accretion as a function of the last T of equilibration in the magma ocean (using our parameterization and Eq. S7 and S8). This has been extensively described in previous works (e.g., Siebert et al., 2013; Badro et al., 2015). All results from the modelling for any P–T–fO₂ conditions fall on this black line. To illustrate our purpose, we picked a selection of P–T–fO₂ conditions (i.e., final P–T condition of in a continuous core formation model, and initial fO₂ conditions) during core formation that can account for the abundances of other siderophile elements such as Ni–Co–V–Cr (all circles on figure 3b). From these P–T–fO₂ conditions, we calculate an amount of Pt in

the mantle after core formation ceased and show that an excess amount of Pt is obtained when geophysical observables (i.e., core composition obtained in light elements at these P–T–fO₂ that matches seismological models) are considered additionally. We understand that there are aspects of the modeling that are difficult to easily understand for readers. This is mainly due to the fact that we tried here to combine our Pt partitioning dataset with state-of-the-art modeling of core formation and test whether the amount of Pt obtained can be consistent with constraints on conditions of core formation and core composition from numerous recent works. To our knowledge, it is the first time that experimental partitioning results for HSE are obtained at these extreme P–T conditions and also put in perspective when considering other observables from other siderophile elements and light elements composition of the core. We believe this modeling provides valuable insights and would like to keep this comprehensive approach. To this purpose, we now give more details and explanations regarding the modeling in the MS and SOM and believe that readers have all materials and references to understand clearly the figures.

I have added many specific comments on the attached annotated pdf file, on both the main text and the supplementary material. For example (lines 62–64), I don't understand why positive ¹⁸²W anomalies should be associated with high HSE contents (the opposite might be expected as Hf/W ratios are elevated by W sequestration in the core).

We agree with the Reviewer that this point is confusing and have now clarified the mention of the ¹⁸²W anomalies in the manuscript. According to some authors chemical core-mantle interactions could lead to high HSE abundances (e.g., Mundl-Petermeier et al., 2020, GCA; Rizo et al., 2019, *Geochem. Persp. Lett*), since the core forms early and is W-rich it should have a negative W anomaly. The ¹⁸²W anomaly literature is now quite diverse; some are adamant these anomalies are due to exchange with the core (e.g., Mundl-Petermeier et al., 2020, GCA) while others invoke other mechanism such as incomplete mantle mixing or early mantle components (e.g., Touboul et al., 2012, *Science*). If the high HSE abundances are not due to core addition, then the anomalies could be positive or negative, depending on the mechanism. **Following from this we have now changed this sentence to say ¹⁸²W heterogeneities instead and added the references from above, three of which were missing from the previous version of the manuscript (lines 62 to 66).**

It also seems odd (lines 827-828) that partial equilibration with impactor cores would lead to lower Pt contents in the mantle than is expected after full equilibration.

This is due to the fact that unequilibrated material is transported directly to the core without interacting with the mantle or leaving a mantle signature. In other words, only a portion of the Pt is re-equilibrated at the high P-T conditions of the Earth's magma ocean that decrease its siderophilicity. Therefore, partial equilibration leads to a lower amount of platinum into the mantle at the end of core formation. **This is now explained more clearly in the supplement (Lines 872 – 884 and 929 – 933).**

Here and elsewhere I may just be misunderstanding what the authors mean to say, but I suggest that they look through these annotated comments carefully because they indicate where the manuscript is not clear.

Thank you for these detailed comments. We have gone through the annotations in the pdf file and made the suggested changes and clarified the points that were not clear.

Laurie Reisberg

Reviewer #3 (Remarks to the Author):

The article is based on an experimental investigation of the metal/silicates partitioning behaviour of platinum (Pt) at high pressure and temperature.

Platinum is found to be significantly less siderophile at high temperatures. Using the experimentally determined Pt metal-silicate partitioning coefficients, it is found that core formation models consistent with geochemical and geophysical constraints predict an excess of Pt in the mantle. This is at odds with the chondritic relative abundances of Highly Siderophile Elements (HSE), which requires that the final Pt mantle content has been set by a late addition of chondritic material (late veneer).

The Pt sequestered in the Mantle during core formation must therefore have been removed by some additional mechanism, which the authors argue could be iron disproportion or sulfide segregation.

The authors also argue that the suprachondritic $^{186}/^{188}$ Os isotopes ratio of Hawaiian magmas could be explained if a Pt-enriched reservoir that did not experience disproportionation could have been preserved.

This is a very nice and interesting paper, with clearly important results, and I believe it is well worth publishing in Nature Communications. I have only relatively minor comments and questions.

> I have several questions and comments concerning the mechanism of Pt removal due to Fe disproportion:

- The estimation of Pt removal due to Fe disproportion seems a bit optimistic to me since the calculation (explained in section 5c of the Methods) assumes that the precipitated Fe droplets equilibrate with the full mantle. If disproportion happens in a magma ocean, then the solid part of the mantle is unlikely to equilibrate with the precipitated iron, and would probably keep its Pt. If instead disproportion happens in the solid part of the mantle, then it is far from obvious that the metal could efficiently segregate from the silicates and reach the core.

I would thus think that disproportion would at most remove the excess Pt from the magma ocean.

If correct, this would significantly affect the amount of Pt removal calculated and presented in figure 4. The Deep Magma Ocean case ($P \sim 65$ GPa) corresponds to a magma ocean of roughly 70% of the total mass of the mantle; if the solid part of the mantle keeps its Pt, then the final Pt concentration should be no less than 30% of its pre-removal concentration. In the Shallow Magma Ocean case, the magma ocean amounts to roughly 35% of the total mass of the mantle,

and the final Pt concentration should be no less than 65% of its pre-removal concentration.

There might be an optimal magma ocean depth: in a shallow magma ocean, the Pt metal/silicate partitioning coefficient is high, but the mass of equilibrated mantle is low; in a deeper magma ocean, the Pt metal/silicate partitioning coefficient is lower, but the mass of equilibrated mantle is higher.

This effect could easily be included in the calculations leading to figure 4. I therefore believe that the authors should either estimate the amount of Pt removal by taking into account partial equilibration of the mantle, or give arguments in support of full mantle equilibration.

Thanks for these interesting comments that stress the complexity of these processes and the multiple variable dependency that needs to be taken into account. Even if we used state-of-the-art core formation modeling, there are still numerous assumptions that are made. Testing the full range of realistic parameters in core formation processes is challenging to say the least and well beyond the scope of the present work. Here, our data and models support the general idea of a Pt excess in the BSE after core formation and we believe this to be a robust first order conclusion. Obviously, changing some parameters or some of our initial conditions could lead to different amounts of Pt leftover. However, within the current frame of constraints that are known from experiments and core formation modeling, we believe that this main conclusion would stay unchanged. For instance, the final depth of the magma ocean has to be deep (i.e., corresponding to pressures above 45 GPa) in order to account for the abundances of moderately volatile elements such as Ni and Co.

More precisely, Reviewer #3 emphasizes on the process of Pt removal from disproportionation. In the present work, we made the assumption of full segregation of disproportionated iron to the core in the magma ocean as well as in the solid lower part of the mantle as proposed in previous works (e.g., Frost et al., 2008; Armstrong et al., 2019). The liquid Fe is then assumed to percolate (or diapir) through the solid lower mantle to merge with the core. If Fe^{2+} also disproportionates in the solid mantle (Frost et al., 2004), the precipitated Fe would be able to sequester some trace elements including Pt. There are different ways by which disproportionated metal could have been lost from the lower mantle. As suggested by most core formation models, core-forming metal likely rained down in a deep magma ocean and ponded at its base before going through the solid lower mantle as diapirs (e.g., Stevenson, 1990; Li & Agee 1996; Rubie et al., 2003; Wade and Wood, 2005; Siebert et al., 2012; Fischer et al., 2015). The descending diapirs could have entrained disproportionated metal as they passed through the lower mantle. Secondly, disproportionated metal can separate by percolative flow to the core as small degree Fe liquids can wet mineral grain boundaries at lower mantle conditions (Takafuji et al., 2004, Shi et al., 2013). Relative chondritic abundances of HSE in the mantle corresponding to late accretion seem to indicate that Pt was efficiently removed by these types of processes. For the sake of clarity, we believe that Figure 4 should keep the same outputs. Including the effect of additional parameters such as a potential inefficient segregation of disproportionated iron in the solid lower mantle is difficult to estimate as the proportion of iron stranded in the lower mantle is currently impossible to estimate and remains speculative.

In the case of inefficient equilibration in the magma ocean more Fe precipitation would be required. For example, 50 % efficiency of equilibration would double the amount of Fe required to remove all the Pt, ~2 wt. % Fe in our case. Other mechanisms such as sulfide segregation could also be invoked as a means of HSE removal. We have now mentioned in the discussion (Line 206) and in the caption for Figure 4a that our calculation assumes efficient mixing. This is also mentioned in the supplement where we give an estimate for of the Fe required for 50 % mixing efficiency (Lines 958 to 962).

-Shouldn't Fe disproportionation also affect the abundance of moderately siderophile elements? The precipitated iron should also carry some of these elements to the core. My guess is that the effect would be small since the metal-silicates partitioning coefficients of these elements isn't very high, but maybe this should be quantified.

At P–T conditions of a deep magma ocean (i.e., directly comparable to conditions of our experiments), the partitioning of moderately siderophile elements (MSE) such as Ni and Co is at least two orders of magnitude lower than Pt. Late accretion accounts only for a few percent of the total mass budget of these elements in the BSE and is often considered negligible to account for the excess of these elements in the mantle. Accordingly, segregation of small amounts of disproportionated metal iron after core formation ceased would have a negligible effect on the final amounts of these elements in the BSE. The abundances of MSE in the BSE are mainly controlled by metal-silicate equilibration in a deep magma ocean during core formation.

- In the model, Fe disproportionation is assumed to happen after core formation. However, Fe disproportionation could start as soon as the pressure in Earth's mantle is high enough, and it should probably be seen as concomitant of core formation. Would this affect the efficiency of Pt removal?

It is possible that iron disproportionation started while core formation was ongoing. However, Pt abundance in the BSE would still be mainly controlled by core-mantle equilibration and the excess of Pt would most likely still be preserved. For this reason, we discuss and quantify only the effect of iron disproportionation after core formation ceased.

> The authors mention sulfide segregation as a possible way of removing the excess Pt, but this is not quantified in the text. Would this mechanism be as efficient as iron exsolution?

Sulfide segregation would be as efficient as a removal mechanism for HSEs. This has been dealt with extensively in previous works (Rubie et al., 2016, Science). Here we consider Fe disproportionation as the mechanism because of the consistency with the post-core formation mantle redox evolution.

REVIEWERS' COMMENTS

Reviewer #1 (Remarks to the Author):

The authors clearly answer to my questions. Experimental details are well explained, and I am convinced with the importance of the study. I recommend it to be published in Nature Communications.

Reviewer #2 (Remarks to the Author):

Review of revised version of Nature Communications ms. 274927, "Reconciling metal-silicate partitioning and late accretion in the Earth", by Suer et al.

I think the revised version of this manuscript is quite clear, and is essentially ready to publish. Some very minor revision is still needed, but this can be done under the guidance of the editor. In my opinion, there is no reason to send this paper out for further review.

My main remaining concern is again related to Figure 3a. The additions to the caption and the main text were quite helpful and I think that I finally understand what this figure is intended to show. However there are some small inconsistencies between the text and the figure that need to be cleared up. In line 162, the BSE Pt content is given as 7.6 ± 1.3 ppb, but in fig 3a this is shown as about 8.5 ± 1.5 ppb. Also, in line 164 the minimum amount of late accreted material required by the HSE constraints is given as 0.34%, but in Fig. 3a the blue line indicates a value of about 0.38%. These differences are small, and both the text and the figure values are within the range of uncertainty. However, inconsistencies of this sort make it very difficult for the reader to follow the model. So it is critical to make sure that the same numbers are used in the text and the figure. This is also true for the supplemental material related to this model (lines 906-913). Finally, what does the dashed blue curve represent in Fig. S4 (basically the same figure as 3a but for partial rather than full equilibrium)? This curve should be explained or removed from the diagram.

Lines 898-903. I was a bit surprised by the statement that Os isotopic signatures support a carbonaceous chondrite composition for the late accreting material. I checked the cited references, and while ref. 71 does indeed argue for a CC component in the late veneer, ref. 6 says "our results are consistent with previous suggestions that the late accreting materials had Re/Os and Pt/Os ratios most similar to H ordinary chondrites or to equilibrated enstatite chondrites". So the Os isotope data are ambiguous on this point. On the other hand, the Ru isotope data that Fischer Godde et al. present in ref. 40 actually do favor a CC composition for the late veneer, in contrast to what they concluded in their earlier paper (ref. 72). As the paragraph goes on to say, none of this matters much for the model. But readers will have more confidence in the discussion if citation errors of this type are avoided.

A few very minor grammatical errors:

Line 731. Change "were" to "was".

Line 763. Change "mostly" to "most".

Line 766. Add "the" before "main metallic blob".

Line 790. Add "the" before "only".

Finally, congratulations to the authors for this important new dataset, and for an intriguing hypothesis that will surely raise a lot of debate.

Reviewer #3 (Remarks to the Author):

I am satisfied by the answers to my comments.

I recommend publication of the paper.

REVIEWERS' COMMENTS

Reviewer #1 and Reviewer #3, recommend the publication of the new version of the manuscript without corrections. We hereafter address shortly the last minor comments raised by Reviewer #2.

Reviewer #1 (Remarks to the Author):

The authors clearly answer to my questions. Experimental details are well explained, and I am convinced with the importance of the study. I recommend it to be published in Nature Communications.

Reviewer #3 (Remarks to the Author):

I am satisfied by the answers to my comments.
I recommend publication of the paper.

Reviewer #2 (Remarks to the Author):

Review of revised version of Nature Communications ms. 274927, "Reconciling metal-silicate partitioning and late accretion in the Earth", by Suer et al.

I think the revised version of this manuscript is quite clear, and is essentially ready to publish. Some very minor revision is still needed, but this can be done under the guidance of the editor. In my opinion, there is no reason to send this paper out for further review.

My main remaining concern is again related to Figure 3a. The additions to the caption and the main text were quite helpful and I think that I finally understand what this figure is intended to show. However there are some small inconsistencies between the text and the figure that need to be cleared up.

-In line 162, the BSE Pt content is given as 7.6 ± 1.3 ppb, but in fig 3a this is shown as about 8.5 ± 1.5 ppb.

Thanks for this remark. There is indeed a small discrepancy in between the BSE value given in the text 7.6 ± 1.3 ppb and the value on Figure 3 (8.6 ± 1.7 ppb). The value displayed on figure 3 is the correct one (determined with a compilation of data using two different methods from Day et al., 2016). The value in the text has now been corrected in accordance to this value.

-Also, in line 164 the minimum amount of late accreted material required by the HSE constraints is given as 0.34%, but in Fig. 3a the blue line indicates a value of about 0.38%.

Thanks for this comment. Again, the correct value is displayed on the figure and the text has been corrected for consistency.

-These differences are small, and both the text and the figure values are within the range of uncertainty. However, inconsistencies of this sort make it very difficult for the reader to follow the model. So it is critical to make sure that the same numbers are used in the text and the figure.

We fully agree with this comment and the text has now been corrected for accuracy.

-This is also true for the supplemental material related to this model (lines 906-913).

This was also changed to 0.38 % M_E for consistency.

Finally, what does

the dashed blue curve represent in Fig. S4 (basically the same figure as 3a but for partial rather than full equilibrium)? This curve should be explained or removed from the diagram.

The dashed blue line is the lower error envelope (derived from uncertainty in the regression model) on the late accretion mass estimates. A sentence has been added to Fig S4 caption.

Lines 898-903. I was a bit surprised by the statement that Os isotopic signatures support a carbonaceous chondrite composition for the late accreting material. I checked the cited references, and while ref. 71 does indeed argue for a CC component in the late veneer, ref. 6 says "our results are consistent with previous suggestions that the late accreting materials had Re/Os and Pt/Os ratios most similar to H ordinary chondrites or to equilibrated enstatite chondrites". So the Os isotope data are ambiguous on this point. On the other hand, the Ru isotope data that Fischer Godde et al. present in ref. 40 actually do favor a CC composition for the late veneer, in contrast to what they concluded in their earlier paper (ref. 72). As the paragraph goes on to say, none of this matters much for the model. But readers will have more confidence in the discussion if citation errors of this type are avoided.

Thanks for bringing up these details about the recent works on the isotopic ratios. This paragraph has now been modified to reflect this comment and the references have now been cited correctly. Ref 6 was removed from the sentence about CC in the late veneer composition and a new sentence inserted to reflect the ambiguity in the interpretations of both the Os and Ru isotopic signatures.

A few very minor grammatical errors:

Line 731. Change "were" to "was".

Line 763. Change "mostly" to "most".

Line 766. Add "the" before "main metallic blob".

Line 790. Add "the" before "only".

These changes have now been made in the supplement.

Finally, congratulations to the authors for this important new dataset, and for an intriguing hypothesis that will surely raise a lot of debate.